# Human-Guided Complexity-Controlled Abstractions

**Andi Peng**[*]
MIT

**Mycal Tucker**[*]
MIT

**Eoin M. Kenny**
MIT

**Noga Zaslavsky**
UC Irvine

**Pulkit Agrawal**
MIT

**Julie A. Shah**
MIT

## Abstract

Neural networks often learn task-specific latent representations that fail to generalize to novel settings or tasks. Conversely, humans learn discrete representations (i.e., concepts or words) at a variety of abstraction levels (e.g., "bird" vs. "sparrow") and deploy the appropriate abstraction based on task. Inspired by this, we train neural models to generate a spectrum of discrete representations and control the complexity of the representations (roughly, how many bits are allocated for encoding inputs) by tuning the entropy of the distribution over representations. In finetuning experiments, using only a small number of labeled examples for a new task, we show that (1) tuning the representation to a task-appropriate complexity level supports the highest finetuning performance, and (2) in a human-participant study, users were able to identify the appropriate complexity level for a downstream task using visualizations of discrete representations. Our results indicate a promising direction for rapid model finetuning by leveraging human insight.

## 1 Introduction

Neural networks learn implicit representations tailored to specific training tasks, but such representations, or abstractions, often fail to generalize to distinct test tasks. One approach to mitigating such generalization failures is based on the Information Bottleneck (IB) method [26, 1, 25, 24], which provides a framework for controlling how much information is passed through the network, effectively limiting its representational complexity. Unfortunately, it is difficult to know *a-priori* how much information to retain for optimal task performance.

Humans, however, when given a task, are remarkably adept at understanding which abstraction to deploy [35]. Consider an expert birdwatcher who has learned fine-grained classes of birds such as "sparrows," "goldfinches," and "kiwis." When accompanied by other experts, a birdwatcher knows to deploy their "most complex" abstraction for classifying birds; meanwhile, when at home with a 6-year old child, a birdwatcher knows to deploy a far simpler representation to communicate about a "red" vs. "yellow" bird. In other words, given a task, humans naturally adopt the right abstraction level that meaningfully captures task-relevant information and enables rapid learning [11, 5]. Inspired by humans, we seek to train neural nets along an axis of representational complexity and allow end users to select the desired complexity level to support data-efficient finetuning.

In this paper, we introduce a human-in-the-loop framework, depicted in Figure 1, for pre-training and finetuning complexity-regulated neural representations. Within this framework, we first generate a spectrum of complexity-controlled representations by training discrete information bottleneck methods [29] on a pre-training task. Second, we allow a human to specify a finetuning task, unknown

---

[*]Equal contribution.
[1]Code available at `github.com/mycal-tucker/human-guided-abstractions`.

37th Conference on Neural Information Processing Systems (NeurIPS 2023).

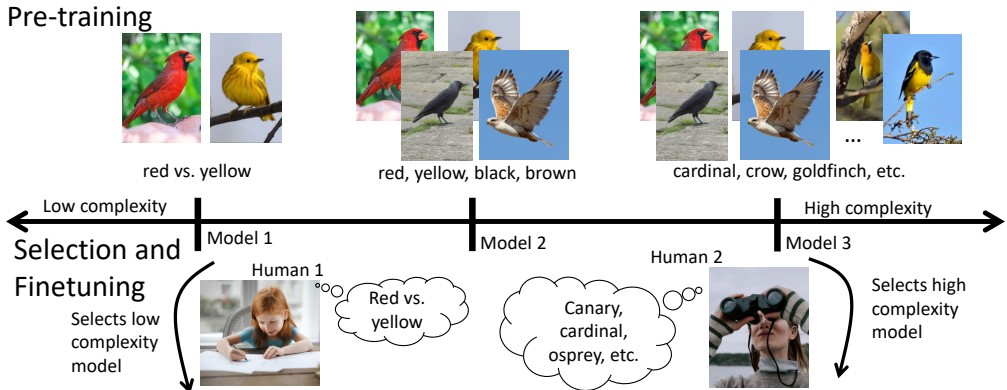

Figure 1: Our human-in-the-loop framework for pre-training and finetuning, illustrated for a bird-identification example. In pre-training, we generate a spectrum of encoders using representations from low to high complexity (e.g., just two crude categories to fine-grained species classifications). In fine-tuning, based on a desired task, a human selects an appropriate model for finetuning (e.g., crude categories for a child learning colors and fine-grained categories for a birder).

*a priori*, and select and finetune a pre-trained model. Given that humans specify the finetuning task, and may have to provide finetuning labels, few-shot adaptation is important.

In computational experiments, we find that finetuning performance is non-monotonically linked to representation complexity: representations that are too complex are data-inefficient, and representations that are too simple fail to capture important information. In a user study, we show that humans, given a desired finetuning task, can select high-performance models from a set of pre-trained models at different complexity levels. For example, as in Figure 1, a child performing a low-complexity task might select a low-complexity representation. More generally, while our computational experiments establish that finetuning performance is a function of model complexity, our study shows that humans can select (near) optimal complexity levels given a task for model finetuning. Lastly, the choice of neural architecture significantly affects finetuning efficiency: on one task, for example, our best-performing architecture, tuned to the right complexity, achieves better performance than other standard encoding methods with $50\times$ more finetuning data.

Our findings suggest that automatically constructing complexity-regulated abstractions during pre-training and then providing a diverse spectrum for human use for finetuning is a promising direction for human-in-the-loop few-shot adaptation. In summary, our contributions are **(1)** introducing a human-in-the-loop framework for automatically generating a spectrum of complexity-regulated abstractions for fast adaptation, **(2)** establishing that finetuning performance is a function of representation complexity, and **(3)** demonstrating the utility of our human-in-the-loop framework in a user study.

## 2 Related Work

### 2.1 Abstraction in Human Cognition

There is substantial evidence that suggests that much of human learning, perception, communication, and cognition may be understood as compression of relevant information [31, 36, 11]. For example, fast human learning can be enabled by merging two or more instances of statistical patterns into one when appropriate [20]. Moreover, the simplicity of these patterns appears key to supporting their predictive power on downstream tasks [2]. This connection between compression and prediction provides an elegant explanation for why human brains have evolved to be such efficient compressors of information and experience [17]. Furthermore, visual abstractions have been found to prioritize functional properties, i.e. downstream task use, at the expense of visual fidelity, i.e. image reconstruction [11]. This suggests that different abstractions are constructed and deployed conditioned on tasks. Inspired by this, we seek to train neural networks to output visual abstractions that are also functionally useful to human users on a diverse range of downstream tasks.

## 2.2 Discrete Information Bottleneck

In our work, we build upon and compare to methods from prior research in discrete information bottlenecks. Generally, such work seeks to generate complexity-limited representations (roughly, limiting the number of bits about the input in a representation) using a finite set of representations (dubbed quantized vectors or prototypes). Recent work [27, 28, 7, 12] has approached this problem by combining ideas from the Variational Information Bottleneck [1, 9] with Vector Quantization (VQ) [29]. In such works, an encoder network outputs parameters to a normal distribution, from which a continuous latent variable is sampled, and the sample is discretized to the closest element of a learnable codebook. By penalizing the KL divergence between the normal and a fixed prior (typically a unit normal), one can limit the complexity of representations. We refer to this family of approaches as the Vector Quantized Variational Information Bottleneck - Normal (VQ-VIB$_\mathcal{N}$). Other work proposes a different sampling mechanism before vector quantization, but experimental evaluation of these methods remains limited [22, 32].

In our work, we propose that complexity-constrained discrete representation learning can be used to generate a meaningful variety of encoders for a human-in-the-loop finetuning process. We use methods from prior work, and we propose a novel combination of entropy regularization and categorical sampling that, in experiments, supports the best finetuning performance.

# 3 Approach

## 3.1 Problem Formulation and Human-in-the-loop Framework

We consider a pre-training and finetuning problem, wherein a model is first trained on a pre-training dataset and must be rapidly adapted to a distinct finetuning task. Unlike classic meta-learning frameworks [6, 21, 19], we do not assume access to a distribution of tasks.

In pre-training, we assume access to a dataset of inputs and pre-training outputs, $(X, Y_p)$ (e.g., images of birds and species labels). That is, $x \sim \mathbb{P}(X)$, $y \sim \mathbb{P}_p(Y|X)$. In finetuning, we assume access to a task-specific dataset with similarly-drawn inputs but novel task labels: $(X, Y_t)$, for $x \sim \mathbb{P}(X)$, $y \sim \mathbb{P}_t(Y|X)$ (e.g., for the depicted girl's finetuning task, the color of the bird). Lastly, we assume that the pre-training labels are a sufficient statistic for the task-specific labels: $I(X; Y_t) = I(Y_p; Y_t)$. Intuitively, this states that the pre-training objective must include relevant information for the finetuning task. This is trivially satisfied with a reconstruction loss (i.e., $Y_p = X$) or fine-grained classifications (e.g., pre-training on exact species, but finetuning on groups of species).

Without information about the finetuning task, it is difficult, *a priori*, to identify how to pre-train a model to perform optimally on the finetuning task. Regularization methods like information bottleneck, for example, depend upon insight on the downstream task to specify the right level of regularization [1]. Rather than automatically identifying the right model, therefore, we propose a human-in-the-loop framework, depicted in Figure 1. Within our framework, we seek to generate a suite of pre-trained neural net encoders such that an end user can identify which one uses abstractions that support high performance for their desired task.

## 3.2 Technical Approach

Within our human-in-the-loop framework, it is critical to generate a "good" set of encoders from which the human chooses. This set must exhibit important variation (such that some encoders are better than others) and human-interpretability (such that humans can select the better encoders). We propose and validate in experiments that complexity-controlled discrete representation learning mechanisms achieve these desiderata. First, discrete representations can support human interpretability (via visualizations of the finite set of representations) [15, 18]. Second, controlling the complexity of representations is a meaningful axis of variation for encoders that likely supports human-specified tasks [34].

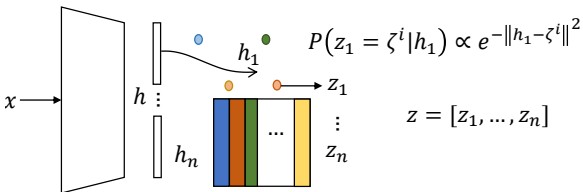

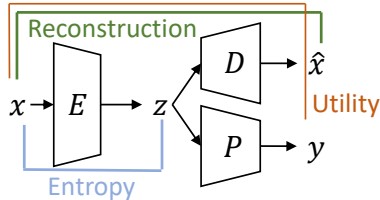

Figure 2: A VQ-VIB$_\mathcal{C}$ encoder maps an input, $x$, to a representation, $h$ in $\mathbb{R}^Z$, which is divided into $n$ sub-representations, $h_i$. Each $h_i$ is discretized stochastically by sampling based on distance to each quantized vector in the codebook, $\zeta$. Lastly, the sampled quantized vectors are concatenated for the full latent representation, $z$.

Figure 3: Different encoders train within an encoder-decoder-predictor framework via utility, entropy, and reconstruction losses [adapted from 27].

## 3.3   Neural Architecture Improvements

To generate a spectrum of encoders using representations at different complexity levels, we extend prior methods in discrete information bottleneck. We briefly present a novel neural method, which we dub the Vector-Quantized Variational Information Bottleneck - Categorical, or VQ-VIB$_\mathcal{C}$.

In VQ-VIB$_\mathcal{C}$, we combine information-theoretic losses from Tucker et al. [27]'s Vector Quantized Variational Information Bottleneck method (which we dub VQ-VIB$_\mathcal{N}$ to emphasize that latent representations are drawn from a normal distribution) with categorical sampling mechanisms similar to Roy et al. [22] and Wu and Flierl [32]. Our VQ-VIB$_\mathcal{C}$ encoder architecture is depicted in Figure 2. The encoder is parametrized by a feature extractor (in our cases, a standard feedforward neural network), an integer, $n$, representing how many quantized vectors to combine into a single latent representation, and $C$ learnable quantized vectors $\zeta^i; i \in [1, C], \zeta^i \in \mathbb{R}^{Z/n}$. Using this architecture, the encoder is characterized as a function mapping from an input to a distribution over latent representations: $q(z|x); x \in \mathbb{R}^X, z \in \mathbb{R}^Z$. Concretely, the VQ-VIB$_\mathcal{C}$ architecture deterministically maps from $x$ to a hidden representation, $h \in \mathbb{R}^Z$. That hidden representation is divided into $n$ vectors of equal size: $h = [h_1, h_2, ..., h_n]$. Each $h_i$ is then probabilistically quantized by sampling a quantized vector according to the L2 distance from $h_i$ to each quantized vector: $\mathbb{P}(z_i = \zeta^j | h_i) \propto e^{-||h_i - \zeta^j||^2}$. (One can differentiate through sampling from this categorical distribution via the gumbel-softmax trick [13, 16].) Lastly, these $n$ sampled discrete representations, each of which is dubbed $z_i$, are concatenated to form the latent representation, $z = [z_1, z_2, ..., z_n]$.

We train VQ-VIB$_\mathcal{C}$ via a combination of losses introduced by Tucker et al. [27]. We assume the VQ-VIB$_\mathcal{C}$ encoder is trained with a decoder and a predictor, as depicted in Figure 3. Given a utility function, $U$, VQ-VIB$_\mathcal{C}$ is trained to maximize the objective function in Equation 1:

$$\max \quad \lambda_U \mathbb{E}[U(x,y)] - \lambda_I \mathbb{E}[||x - \hat{x}||^2] - \lambda_H \mathbb{E}\left[ \sum_{i \in [1,n]} H(\mathbb{P}(\zeta | h_i(x))) \right]$$
$$- ||\mathtt{sg}[h_i(x)] - \zeta_i(x)||^2 - \alpha ||h_i(x) - \mathtt{sg}[\zeta_i(x)]||^2 \tag{1}$$

This objective trades off, in order in the equation: 1) maximizing the expected utility (e.g., cross-entropy loss), 2) minimizing the expected reconstruction loss (here, MSE), 3) minimizing the entropy of the distribution over codebook elements, and 4) minimizing a clustering loss from prior literature, encouraging encodings and quantized vectors to cluster (and, as in prior art, we leave $\alpha = 0.25$ in all experiments) [29]. Here, $\mathtt{sg}$ represents the stop-gradient operation.

We include a more thorough discussion of this loss function, and comparisons to losses and architectures proposed in prior literature, in Appendix A. We emphasize that, while we propose some modifications to neural architectures, the primary contributions of this work are not about VQ-VIB$_\mathcal{C}$ but rather: 1) our human-in-the-loop framework and 2) the recognition that discrete information bottleneck methods support high performance within this framework. In experiments, we found that our VQ-VIB$_\mathcal{C}$ method performed better than methods from prior art, so we include this technical section to inform future researchers about modest architectural changes that support better performance.

# 4 Assessing Representational Complexity's Impact on Finetuning

We first present results for computational experiments in three different visual classification domains, indicating that finetuning performance varies as a function of representation complexity in a non-monotonic way. All experiments consisted of pre-training and finetuning phases. First, in pre-training on a low-level task, we generated a spectrum of models at varying complexity levels by varying loss hyperparameters. Second, we used the encoders from the pre-trained models and trained predictors to map from encodings to downstream predictions. Jointly, these steps enabled us to assess the importance of complexity-controlled representations for data-efficient finetuning. Overall, we found that, for small amounts of finetuning data, tuning representations to the right complexity was important, but with large amounts of data, any sufficiently-complex encoder performed similarly.

## 4.1 Domains

We trained agents on three image classification datasets: FashionMNIST [33], CIFAR100 [14], and iNaturalist 2019 (iNat) [10]. For FashionMNIST, we used a two-level hierarchy, grouping the 10 low-level classes into 3 higher-level classes (`top` = Tshirt/top, Pullover, Coat, and Shirt; `shoes` = Sandal, Sneaker, and Ankle Boot; `other` = Trouser, Dress, Bag). The CIFAR100 and iNat datasets have more extensive hierarchical structures, as detailed by, e.g., Sainte Fare Garnot and Landrieu [23]. For both datasets, we considered two levels of increasing crudeness in the hierarchy. For CIFAR100, we used both a 20-way division (each class consisting of 5 low-level classes) and a binary division for alive vs. non-alive objects. For iNat, we used a 34-way division (representing categories like "butterflies" and "mushrooms") and a 3-way division for plants, animals, or fungi. Thus, all domains were characterized by semantically-meaningful hierarchies.

## 4.2 Pretraining and Finetuning

Our experiments comprised two phases: pre-training and finetuning. In pre-training, we trained an encoder, decoder, and predictor on a low-level task. For example, for iNat, this consisted of classifying a photograph among 1010 species. During pre-training, after convergence to high accuracy, we decreased the complexity of representations by tuning a hyperparameter until representations were uninformative. For our $\beta$-VAE and VQ-VIB$_\mathcal{N}$ baselines, we increased $\lambda_C$, a scalar weight penalizing the KL divergence of the conditional Gaussian (generated by the encoder) from a unit Gaussian. (See Appendix A for details of training losses for $\beta$-VAE and VQ-VIB$_\mathcal{N}$, including $\lambda_C$. In general, larger values of $\lambda_C$ led to less complex representations and greater MSE values for reconstructions.) For VQ-VIB$_\mathcal{C}$, we increased $\lambda_H$, the scalar weight penalizing the entropy of the categorical distribution over quantized vectors. In Appendix D, we examine other combinations of losses and architectures to control complexity, but found that tuning entropy for VQ-VIB$_\mathcal{C}$ achieved the best results. In general, to address some conflicting definitions of complexity in prior literature, we distinguish between an entropy loss, as introduced for our VQ-VIB$_\mathcal{C}$ model, and a complexity loss, where we use the definition of complexity as $I(X; Z)$.

In finetuning, we trained new predictor networks to map from encodings to classifications, using a small amount of training data. Using an encoder saved during pre-training, we generated encodings from inputs. New predictors were trained on classification tasks given the supervised (encoding, label) data. We generated $k$ encodings for each class in the supervised dataset, and report results for different $k$. By varying $k$ and which encoder was used to generate encodings (from high to low complexity), we could investigate the effect of encoder complexity on data-efficiency in finetuning. In all experiments, we pre-trained 5 models and ran 10 finetuning trials for each encoder. Further details on pre-training and finetuning are included in Appendix A.

## 4.3 Illustrative Example: FashionMNIST

We illustrate the high-level trends from our computational results using the FashionMNIST dataset. We trained models on the 10-way classification task to high accuracy (typically around 90%) and decreased representational complexity until the model was accurate 10% of the time (random chance).

Depictions of the learned quantized vectors for VQ-VIB$_\mathcal{C}$ are included in Figure 4 at high and low complexities. At high complexity, VQ-VIB$_\mathcal{C}$ used a large number of distinct prototypes, including multiple prototypes per class (e.g., for two different types of handbags). This high complexity

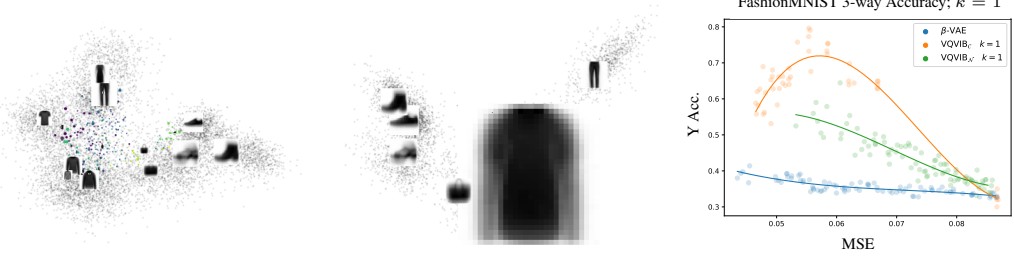

(a) VQ-VIB$_\mathcal{C}$; high complexity      (b) VQ-VIB$_\mathcal{C}$; low complexity      (c) Finetuning accuracy vs. MSE.

Figure 4: 2D PCA of VQ-VIB$_\mathcal{C}$ latent representations for FashionMNIST at high (a) or low (b) complexity. Colorful points and images are prototypes, scaled in proportion to frequency of use. At high complexity, VQ-VIB$_\mathcal{C}$ uses a large number of prototypes, including multiple per class (e.g., two types of handbags); at low complexity, classes merge (e.g. different types of shirts). c) When finetuned on a 3-way classification task with 1 example per class, VQ-VIB$_\mathcal{C}$ outperformed $\beta$-VAE and VQ-VIB$_\mathcal{N}$, and less complex representations (greater MSE) improved model scores.

supported low MSE reconstruction loss. As we decreased the representational complexity, VQ-VIB$_\mathcal{C}$ used fewer prototypes that represented more abstract concepts (Figure 4 b). For example, five distinct classes (Tshirt/top, Pullover, Coat, Shirt, and Dress) were merged into a single prototype, increasing MSE. Further examples of prototype evolution are in Appendix C.

In finetuning experiments, we loaded VQ-VIB$_\mathcal{C}$ encoders across a range of complexities and finetuned a predictor on the 3-way classification task described earlier (tops, shoes, or other). Using just one example per class ($k = 1$), VQ-VIB$_\mathcal{C}$ models were more accurate than $\beta$-VAE or VQ-VIB$_\mathcal{N}$ models, as shown in Figure 4 c. The $x$ axis displays each encoder's MSE, which is a proxy for the inverse of the complexity of representations: more complex representations capture more information and generate better reconstructions (lower MSE). The $y$ axis shows the finetuned predictor's accuracy on the 3-way task. While VQ-VIB$_\mathcal{C}$ peak performance is roughly 70% accuracy (remarkably high, given $k = 1$), $\beta$-VAE and VQ-VIB$_\mathcal{N}$ performance peaks at 40% and 55%, respectively.

Beyond comparing VQ-VIB$_\mathcal{C}$ to other models, Figure 4 c shows the importance of task-appropriate complexity levels. At low MSE values, VQ-VIB$_\mathcal{C}$ uses many different prototypes, which impedes finetuning data efficiency. However, as VQ-VIB$_\mathcal{C}$ learns more abstract representations (which increases MSE), finetuning performance improves (the correlation between MSE and accuracy is positive for MSE $< 0.055$   ($p < 0.03$)). However, past an MSE value around 0.055, VQ-VIB$_\mathcal{C}$ lacks the representational capacity to distinguish between images that should be classified differently, and performance worsens.

Here, we discussed finetuning results for 1 labeled example per class ($k = 1$) and for $n = 1$, the number of quantized vectors to combine into representations, but results and analysis for $k \in [1, 2, 5, 10, 50]$, $n \in [1, 2, 4]$, and for $\beta$-VAE, VQ-VIB$_\mathcal{N}$, and VQ-VIB$_\mathcal{C}$ models are included in Appendix B. For all $k$, we found that VQ-VIB$_\mathcal{C}$ outperformed $\beta$-VAE and VQ-VIB$_\mathcal{N}$. In fact, VQ-VIB$_\mathcal{C}$ with $k = 1$ outperformed $\beta$-VAE for $k = 50$, indicating important architectural benefits. Increasing $n$ supported higher complexity (lower MSE) but worse fine-tuning performance; intuitively, many discrete representations approximate continuous representations, which have worse sample complexity. This is an important limitation of combinatorial codebooks that others propose [28, 12].

A series of ablation studies confirm the importance of the VQ-VIB$_\mathcal{C}$ architecture and annealing entropy, instead of complexity, for efficient finetuning (Appendix D, for FashionMNIST and other domains). In particular, we found that for VQ-VIB$_\mathcal{C}$ and VQ-VIB$_\mathcal{N}$, penalizing entropy led to greater finetuning accuracy than when penalizing complexity, and VQ-VIB$_\mathcal{C}$ outperformed VQ-VIB$_\mathcal{N}$ when both were trained via entropy regularization. In other words, penalizing entropy enabled better finetuning performance, and VQ-VIB$_\mathcal{C}$ supported better entropy regularization.

Lastly, given the importance of complexity on finetuning performance, we considered two methods for autonomously selecting the optimal complexity level. For low-data regimes ($k \leq 5$), the simple heuristic of choosing the most complex encoder was clearly suboptimal; however, as the amount of finetuning data increased (e.g., $k = 50$), the most complex encoders achieved near-optimal

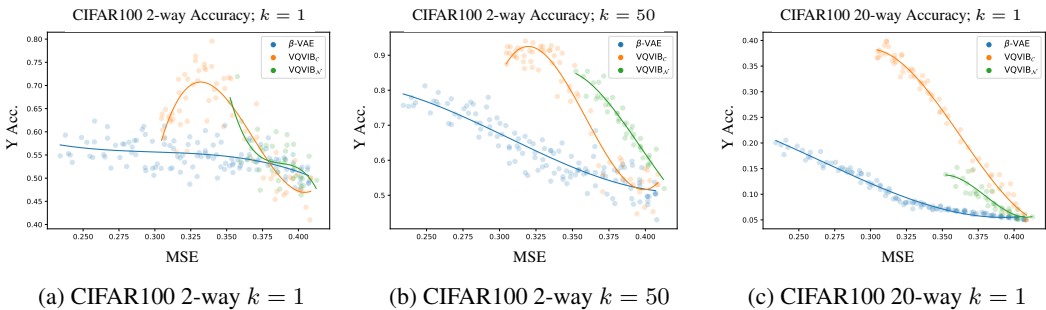

(a) CIFAR100 2-way $k = 1$     (b) CIFAR100 2-way $k = 50$     (c) CIFAR100 20-way $k = 1$

Figure 5: CIFAR100 finetuning, using 1 (a) or 50 (b) training examples for living vs. non-living things. With small amounts of data VQ-VIB$_\mathcal{C}$ benefits from tuning to the right complexity level. When finetuning on the 20-way classification task (c) VQ-VIB$_\mathcal{C}$ continues to outperform other architectures.

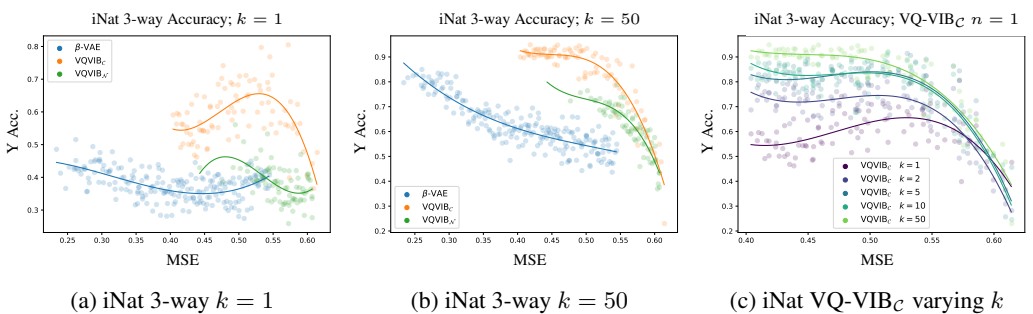

(a) iNat 3-way $k = 1$     (b) iNat 3-way $k = 50$     (c) iNat VQ-VIB$_\mathcal{C}$ varying $k$

Figure 6: iNat finetuning results for the plant-animal-fungus classification task. Using only 1 finetuning example (a), VQ-VIB$_\mathcal{C}$ performance improved as complexity decreased, and outperformed $\beta$-VAE and VQ-VIB$_\mathcal{N}$. Using 50 finetuning examples, performance plateaued, but VQ-VIB$_\mathcal{C}$ continued to outperform other methods (b). VQ-VIB$_\mathcal{C}$ performance shifts smoothly for varying $k$ (c).

performance (see Figure 10 in Appendix B). We also tested methods for selecting encoders via validation-set accuracy and found similar trends. In finetuning, we held out a subset of the data for assessing finetuning accuracy and selected the best-performing encoder via validation set accuracy. For $k \leq 5$, this validation-set approach did not consistently converge to optimal performance, but, for sufficiently large $k$, it did. Further results from this approach are included in Table 4 and Appendix B. Generally, we found that for large enough $k$, the importance of tuning to the right complexity decreases, and several methods can select optimal encoders; for very small $k$, however, autonomous methods are suboptimal.

This illustrative FashionMNIST use case demonstrates many of the important trends we explore in our later experiments: tuning VQ-VIB$_\mathcal{C}$ representations to the "right" complexity was important for optimal performance for small $k$, and VQ-VIB$_\mathcal{C}$ generally outperformed other encoders for few-shot finetuning.

### 4.4 CIFAR100 and iNat

In this section, we present results from computational experiments in the more challenging CIFAR100 and iNat domains. As before, in the smallest data regime (small $k$), using less complex representations afforded greater efficiency benefits, and VQ-VIB$_\mathcal{C}$ outperformed $\beta$-VAE and VQ-VIB$_\mathcal{N}$.

Figure 5 a and b show finetuning accuracy on the binary classification task for CIFAR100 on identifying living vs. non-living things. With only 1 datapoint per class (Figure 5 a), $\beta$-VAE performance remained effectively flat at random chance: barely exceeding 50%. However, for VQ-VIB$_\mathcal{C}$, for MSE < 0.35, performance increased as MSE increased (linear regression slope was positive $p < 0.001$), up to a 70% accuracy rate, before worsening as MSE increased further. Increasing $k$ to 50 (Figure 5 b) shrank the gap between the encoder architectures, and flattened the improvements previously observed, but VQ-VIB$_\mathcal{C}$ continued to outperform $\beta$-VAE and VQ-

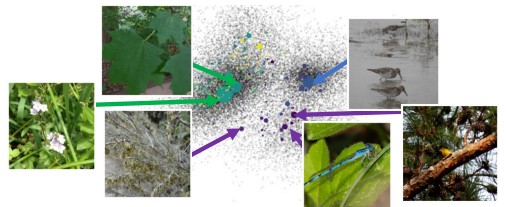 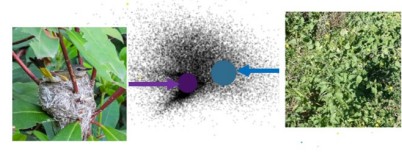

(a) PCA of high-complexity VQ-VIB$_\mathcal{C}$. Note the orange bird in the pine tree.

(b) PCA of low-complexity VQ-VIB$_\mathcal{C}$. Note the bird in its nest on the left.

Figure 7: 2D PCA of the VQ-VIB$_\mathcal{C}$ latent space for iNat, at different complexity levels. Each gray point represents the continuous encoding output by the encoder; the colorful points represent the learnable prototypes. For visualization purposes, the closest training image to each prototype is included in the diagrams. In the high-complexity case, the model has learned many distinct prototypes, representing concepts like birds or insects. In the low-complexity case, VQ-VIB$_\mathcal{C}$ uses only two prototypes, which capture a plant-animal distinction the aligns closely with the iNat hierarchy.

VIB$_\mathcal{N}$ models. When finetuned on the 20-way classification task, the same trends of VQ-VIB$_\mathcal{C}$ outperforming other architectures held (Figure 5 c), although the finetuning accuracy decreased monotonically as MSE increased, rather than peaking at a specific complexity. Plots for a larger range of $k$, and for varying $n$, for both finetuning tasks, are included in Appendix B.

Similar trends held in the iNat dataset when finetuned on the 3-way animal-plant-fungus task, as depicted in Figure 6. For small $k$, VQ-VIB$_\mathcal{C}$ finetuning performance initially improved as the encoder learned more compressed representations (Figure 6 a). For VQ-VIB$_\mathcal{C}$, $k = 1$, the linear correlation between MSE and finetuning accuracy is positive ($p < 0.001$) for MSE $< 0.55$. Further analysis of finetuning performance for VQ-VIB$_\mathcal{C}$, for varying $k$, shows a smooth change in behavior as $k$ increases: simultaneously improving overall performance, and benefiting less from compressed representations (Figure 6 c). This indicates that VQ-VIB$_\mathcal{C}$ is most advantageous in a low-data regime. Similar results for finetuning on the 34-way finetuning task are included in Appendix B. Visualization via 2D principle component analysis (PCA) confirms our intuition of how VQ-VIB$_\mathcal{C}$ models support few-shot learning for iNat. At lower complexity levels, a VQ-VIB$_\mathcal{C}$ encoder used a decreasing number of prototypes to represent increasingly abstract concepts (Figure 7).

As in the FashionMNIST domain, we evaluated autonomous methods for selecting the optimal encoder and found that, while they succeeded for sufficiently large $k$, they struggled in a low-data regime. For $k = 1$, for example, the heuristic of choosing the most complex encoder was suboptimal (e.g., see Figure 5 a and Figure 6 a). At the same time, for $k = 50$, this heuristic worked quite well (e.g., Figure 5 b and Figure 6 b). Lastly, selecting models via validation set accuracy similarly worked well for large enough $k$, but struggled for $k \leq 5$ (see Appendix B).

Finally, we briefly note some limitations of VQ-VIB$_\mathcal{C}$ and our finetuning method. In the results discussed so far, we found consistent advantages in using VQ-VIB$_\mathcal{C}$ and low-complexity representations. However, as the amount of finetuning data increases, more complex representation methods like $\beta$-VAEs outperform VQ-VIB$_\mathcal{C}$. A more complete discussion of this phenomenon is included in Appendix B. Between our main results and these limitations, we find that the data-efficiency benefits of using VQ-VIB$_\mathcal{C}$ in finetuning are greatest in data-poor and low-complexity settings.

## 5 Human-in-the-Loop Selection of Task-Appropriate Representations

Given our main motivation of a human-in-the-loop framework for selecting task-appropriate abstractions (recall Figure 1), we conducted a human-participant study to evaluate whether users could select the highest-performing task-appropriate models given visualizations of prototypes as task abstractions. This is important because, in order to take full advantage of our setup, users must be able to select the appropriate task-level representation so the neural network best learns the finetuning task in a sample efficient manner. We asked users to select the optimal encoder for a given task, given a visualization of encoder prototypes. A positive result would show that, given a spectrum of pre-trained VQ-VIB$_\mathcal{C}$ encoders, a human user wishing to deploy a task-appropriate model could specify the right encoder for a given task, and thus fine-tune the model in a sample efficient manner.

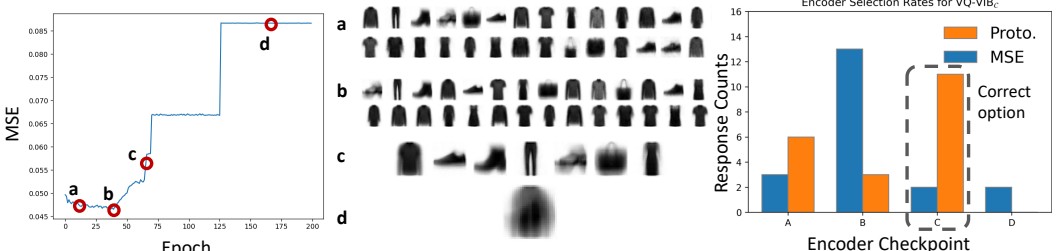

Figure 8: Participants were asked to select the optimal encoder for the FashionMNIST finetuning task based on MSE values (left) or decoded representations (middle). Using only MSE values, users often incorrectly selected option b), whereas when viewing prototypes, users were able to correctly identify the appropriate abstraction level, and thus the optimal encoder (c).

## 5.1 User Study

Our human experiment was performed using the pre-trained models from the FashionMNIST domain in Section 4.3. Users were told that a robot was good at sorting clothing into 10 distinct categories (i.e., the 10 normal categories in FashionMNIST), but that in their case they should consider a different set of categories. Each user was randomly assigned a model type (VQ-VIB$_{\mathcal{C}}$ or VQ-VIB$_{\mathcal{N}}$) and a visualization type (prototypes or a plot of MSE during training). In three questions (explained below), users were shown different groupings of the 10 FashionMNIST classes and asked to select which of four model checkpoints they thought best represented the groupings. Overall, this between-subjects setup allowed us to compare the effects of different visualization methods on user accuracy in selecting the optimal representation.

For our main question, in which users were asked to select an encoder that could sort FashionMNIST items into the three finetuning categories (tops, shoes, and other), our null hypothesis, $\mathbf{H_0}$, was *"Users viewing VQ-VIB$_{\mathcal{C}}$ prototypes will select the optimal encoder as often as those viewing MSE scores".* Our alternative hypothesis, $\mathbf{H_1}$, was *"Viewing VQ-VIB$_{\mathcal{C}}$ prototypes improves users' ability to select the optimal encoder compared to just viewing MSE scores."*

**Participants.** We crowd-sourced 20 participants per group from Prolific.com (N=80). The sample was balanced to have an even number of male and female participants. All participants were native English speakers above the age of 18 and resided in the U.S., the U.K., or Ireland. Given estimates of survey duration, calculated in a pilot study, we paid participants according to an estimated $12 per hour wage. The study received IRB approval from MIT.

**Materials.** Each user was presented three questions, corresponding to different groupings of the 10 FashionMNIST classes: in Question 1 classes were grouped according to the finetuning labels in our computational experiments (tops, shoes, and other), in Question 2, there were 10 distinct classes corresponding to the 10 FashionMNIST classes,and in Question 3, classes were grouped according to an unintuitive 3-way grouping (e.g., Pullover, Dress, Sneaker were one class). Despite the numbering, we randomized the order of questions for each participant. In each question, users were asked to select which of four model checkpoints they thought best represented groupings for that specific question. The four models corresponded to checkpoints taken 1) near the start of training, 2) at the minimum MSE value, 3) at an intermediate MSE value, and 4) at the end of training. We primarily focused on results for Question 1, as it was the only question for which selecting the most complex encoder was not optimal. Responses were categorized as correct if users picked the optimal encoder (option "c" in Figure 8) and otherwise incorrect (but during the study, users were not told whether they were correct).

## 5.2 Results

Results from our user study supported our hypothesis. As shown in Figure 8, for Question 1, corresponding to the 3-way grouping from our computational experiments, users who viewed VQ-VIB$_{\mathcal{C}}$ prototypes selected the optimal encoder 55% of the time, compared to 10% accuracy when

viewing MSE scores (prototype accuracy was significantly greater at $p < 0.01$ for a Fisher's exact test). Interestingly, users who viewed VQ-VIB$_\mathcal{N}$ models never achieved high performance (binomial test for non-random chance at $p > 0.2$), suggesting an important advantage of the VQ-VIB$_\mathcal{C}$ architecture. Thus, VQ-VIB$_\mathcal{C}$ prototypes was the *only* visualization method that supported greater-than-random chance performance in selecting optimal abstractions.

Further analysis, using responses to Questions 2 and 3 as well, highlighted the benefit afforded by VQ-VIB$_\mathcal{C}$ prototypes. Overall, we observed high accuracy rates for Questions 2 and 3 regardless of visualization method; this is unsurprising given that the correct behavior for both questions was the selecting most complex encoder. We fitted a Mixed Linear Effects Model (MLEM) to the survey results, predicting user accuracy as a function of model visualization and question, grouped by participant. Using Wilkinson notation [30], our model was: $Acc. \sim Q + M + Q : M + (1|Participant)$ where $Q$ represents the categorical variable for question and $M$ represents the categorical variable for model type and visualization (VQ-VIB$_\mathcal{C}$ Prototypes, VQ-VIB$_\mathcal{C}$ MSE, VQ-VIB$_\mathcal{N}$ Prototypes, etc.). We grouped results by participant to model random-intercept effects of some participants being more accurate than others. Full results for accuracy rates and the MLEM model are included in Appendix E.

Overall, we found a significant ($p < 0.05$) positive interaction effect between Question 1 and visualization of VQ-VIB$_\mathcal{C}$ prototypes. The only other significant effect we found was a negative effect for Question 1. Together, these results show that 1) Question 1 was harder than the other two questions and 2) viewing VQ-VIB$_\mathcal{C}$ prototypes mitigated the effects of increased difficulty for Question 1. This provides crucial support for our hypothesis, showing that viewing VQ-VIB$_\mathcal{C}$ prototypes enabled participants to select the optimal encoder more often.

Lastly, the combination of all results helps rule out several alternative hypotheses for how participants selected encoders based on VQ-VIB$_\mathcal{C}$ prototypes. Users did not merely select the most complex encoder; otherwise, they would have performed poorly on Question 1. Similarly, users did not simply use the fewest number of prototypes greater than the number of finetuning classes; otherwise, they would have performed worse on Question 3. Thus, our results suggest that users interpreted VQ-VIB$_\mathcal{C}$ prototypes as desired in intelligently selecting task-appropriate representations.

# 6 Contributions

We proposed a human-in-the-loop framework for selecting task-appropriate representations for data-efficient adaptation to new tasks. Discrete information bottleneck methods provide a principled way to generate representations along a spectrum of low to high complexity; we tested methods from prior literature and a novel architecture, VQ-VIB$_\mathcal{C}$, for generating such representations.

We found that controlling the complexity of representations was important for data-efficient fine-tuning: overly-complex representations required more training examples, but overly-simplified representations failed to capture important information. In computational studies, we showed the importance of learning low-entropy representations and that VQ-VIB$_\mathcal{C}$ tends to outperform other discrete information bottleneck methods. In a human study, we found that human partners are able to identify optimal complexity levels, indicating a promising direction for human-in-the-loop training.

**Broader Impact:** Generally, we hope that this work supports broader and better human-AI collaboration, supporting human-selected individualized models for particular use cases; however, we recognize that our current prototype-inspection framework may be limited and that care must be taken in future work to verify that representations along the complexity spectrum correspond to desired human concepts. In future work, we hope to explore models that simultaneously support different complexity levels instead of training separate models for different levels.

# 7 Acknowledgements

We thank members of the Interactive Robotics Group for helpful feedback and discussions. Andi Peng is supported by the NSF Graduate Research Fellowship and Open Philanthropy. Mycal Tucker is supported by an Amazon Alexa Science Hub Fellowship.

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

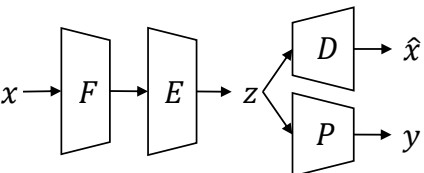

Figure 9: In experiments, we used a common feature-extractor ($F$), decoder ($D$), and predictor ($P$) network backbone while replacing different encoder heads ($E$).

## A Implementation details

### A.1 Pretraining

In pre-training (before the finetuning with a small number of examples on a cruder task), we used the following setup. In general, the overall network architecture comprised a feature extractor, an encoder head, a decoder, and a predictor, as depicted in Figure 9.

In all experiments, the predictor was parametrized as a two-layer feedforward neural network with hidden dimension 128 and a ReLU activation. The last layer's dimension depended upon the exact prediction task (e.g., 10 neurons for FashionMNIST, 100 for CIFAR100, and 1010 for iNat) and used a softmax activation.

The feature extractors and decoders varied by domain. For FashionMNIST, the feature extractor used 3 2D convolution layers, followed by one fully connected layer. The decoder used two linear layers, followed by 3 inverse convolution layers. We again emphasize that the code for these models is available in our codebase, linked to at the beginning of this section.

For CIFAR100 and iNat, we pre-processed the images to extract the 512-dimensional activations from the penultimate layer of a ResNet18 pretrained on ImageNet [8]. These features were used as inputs to the feature extractor ($x$ in Figure 9). For both CIFAR100 and iNat, the feature extractor used two linear layers, with a ReLU activation after the first layer, which had hidden dimension 128. The decoder was used to reconstruct the 512-dimension outputs of the ResNet18, using 3 fully-connected layers of dimension 128, 256, and 512, with ReLU activations between layers.

The different encoder heads were $\beta$-VAE, VQ-VIB$_{\mathcal{C}}$, and VQ-VIB$_{\mathcal{N}}$ models. $\beta$-VAE models used two linear layers, branching off the output of the feature extractor, to generate $\mu$ and $\sigma$ from which to sample a continuous latent variable. VQ-VIB$_{\mathcal{C}}$ directly passed the output of the feature extractor into the vector quantization layer, from which the discrete latent representations were sampled, as described in the main paper. In VQ-VIB$_{\mathcal{N}}$, the output of the feature extractor was passed through two linear layers to generate a $\mu$ and a $\sigma$ (exactly as in the $\beta$-VAE case) before the sampled continuous representation was discretized via vector quantization. Across experiments, the only differences among encoder heads that could arise were due to different latent dimensions (although we fixed it to 32 for all experiments) or, for VQ-VIB$_{\mathcal{C}}$ and VQ-VIB$_{\mathcal{N}}$, the number of elements in the learnable codebook or $n$, the number of quantized vectors to combine into a latent representation.

In the main paper, we discussed the VQ-VIB$_{\mathcal{C}}$ training loss (Equation 1), maximizing utility, minimizing reconstruction loss, and minimizing the entropy of the categorical distribution over codebook elements. A strict generalization of Equation 1, in which a variational bound on the complexity of representations is also penalized, is included in Equation 2:

$$
\begin{aligned}
\max \quad & \lambda_U \mathbb{E}[U(x,y)] - \lambda_I \mathbb{E}[\|x - \hat{x}\|^2] \\
& - \lambda_H \mathbb{E}\left[ \sum_{i \in [1,n]} H(\mathbb{P}(\zeta|h_i(x)) \right] \\
& - \lambda_C \mathbb{E}\left[ D_{\mathrm{KL}}[\mathbb{P}(\zeta|h(x))\|\mathcal{U}(C)] \right] \\
& - \|\mathbf{sg}[h_i(x)] - \zeta_i(x)\|^2 - \alpha\|h_i(x) - \mathbf{sg}[\zeta_i(x)]\|^2
\end{aligned}
\tag{2}
$$

Equation 2 differs from Equation 1 via the third line, penalizing the KL divergence between the conditional categorical distribution over codebook elements and a uniform distribution over the $C$ elements. This provides a variational bound on $I(X, Z)$, dubbed the complexity of representations in prior literature [36, 27]. In our main experiments, we set $\lambda_C = 0$ and vary $\lambda_H$; ablation studies in which we varied $\lambda_C$ instead of $\lambda_H$ are included in Appendix D and confirm that controlling the entropy of representations supported better finetuning accuracy.

In training $\beta$-VAEs, we trained to maximize the function described in Equation 3, where $\mu(x)$ and $\sigma(x)$ represent the $\mu$ and $\sigma$ parameters output by the encoder.

$$
\begin{aligned}
\max \quad & \lambda_U \mathbb{E}[U(x,y)] - \lambda_I \mathbb{E}[\|x - \hat{x}\|^2] \\
& - \lambda_C \mathbb{E}\left[D_{\text{KL}}[\mathcal{N}(\mu(x), \sigma(x))\|\mathcal{N}(0,1)]\right]
\end{aligned}
\tag{3}
$$

Equation 3 trains agents to maximize classification accuracy, minimize MSE, and minimize the complexity of representations. The scalar weight $\lambda_C$ can be viewed as a Lagrange multiplier, constraining how much information can be encoded in representations. This equation is closely related to Equation 2 but, given the continuous nature of encodings in $\beta$-VAE, we could not penalize the entropy of a categorical distribution.

In training VQ-VIB$_{\mathcal{N}}$, we used the training objective proposed by Tucker et al. [27], which closely resembles the training loss for VQ-VIB$_{\mathcal{C}}$ and is shown in Equation 4 (and closely matches Equation 2). We use the same notation as for the $\beta$-VAE and VQ-VIB$_{\mathcal{C}}$ models.

$$
\begin{aligned}
\max \quad & \lambda_U \mathbb{E}[U(x,y)] - \lambda_I \mathbb{E}[\|x - \hat{x}\|^2] \\
& - \lambda_H \mathbb{E}\left[\hat{H}(\mathbb{P}(\zeta|\mu(x))\right] \\
& - \lambda_C \mathbb{E}\left[D_{\text{KL}}[\mathcal{N}(\mu(x), \sigma(x))\|\mathcal{N}(0,1)]\right] \\
& - \|\text{sg}[h_i(x)] - \zeta_i(x)\|^2 - \alpha\|h_i(x) - \text{sg}[\zeta_i(x)]\|^2
\end{aligned}
\tag{4}
$$

The two key differences between Equation 4 and Equation 2 (used for training VQ-VIB$_{\mathcal{C}}$) are bounds on entropy and complexity (on the second and third lines of Equation 4). Just as for $\beta$-VAEs, VQ-VIB$_{\mathcal{N}}$ models uses a KL divergence loss to regulate the complexity of representations. Increasing $\lambda_C$, as we did while annealing complexity in experiments, decreases the amount of encoded information. However, as shown in our results, simply increasing $\lambda_C$ does not ensure that VQ-VIB$_{\mathcal{N}}$ models will use fewer discrete representations. (For visualizations of this effect, see Appendix C.) Tucker et al. [27] advocate for using a small positive $\lambda_H$ to penalize the estimated entropy over codebook elements. We explore varying $\lambda_H$ in Appendix D and find some benefits relative to our main experiments, in which we set $\lambda_H = 0$. However, given that VQ-VIB$_{\mathcal{N}}$ only supports an *approximation* of the entropy term, we find that controlling the entropy for VQ-VIB$_{\mathcal{N}}$ is not as effective as controlling the entropy for VQ-VIB$_{\mathcal{C}}$.

## A.2    Finetuning

In finetuning, we loaded pretrained frozen encoders and trained new predictor models to map from encodings to downstream predictions.

For a finetuning task with $M$ distinct classes, and a "duplication factor," $k$, for how many examples of each class to train with, we randomly selected $k * M$ datapoints to train with. For example, when finetuning on the binary CIFAR100 task of living vs. non-living things, $M = 2$, so we loaded 2 total datapoints for $k = 1$, 4 datapoints for $k = 2$, etc.. For each input in the finetuning dataset, we generated an encoding by passing through the encoder once. This generated a new dataset of encodings and labels, which we used the train the predictor. (Note that this approach is distinct from passing the input through the encoder many times; given stochastic encoders, which we used, the same input could result in many different encodings. Here, we assumed a limited budget of *encodings*.)

Predictor neural networks were instantiated as feedforward networks with 4 fully connected layers, with hidden dimension 256 and ReLU activations, and trained to map from shifted encodings (see

Table 1: Hyperparameters for FashionMNIST training.

| ENCODER | LATENT DIM | $C$ | $n$ | $\lambda_U$ | $\lambda_I$ | $\lambda_{C_0}$ | $\lambda_C$ INCR | $\lambda_{H_0}$ | $\lambda_H$ INCR |
|---|---|---|---|---|---|---|---|---|---|
| $\beta$-VAE | 32 | NA | NA | 10 | 10 | 0.01 | 0.5 | 0.0 | 0.0 |
| VQ-VIB$_\mathcal{N}$ | 32 | 1000 | 1 | 10 | 10 | 0.01 | 0.5 | 0.0 | 0.0 |
| VQ-VIB$_\mathcal{N}$ | 32 | 1000 | 2 | 10 | 10 | 0.01 | 0.5 | 0.0 | 0.0 |
| VQ-VIB$_\mathcal{N}$ | 32 | 1000 | 4 | 10 | 10 | 0.01 | 0.5 | 0.0 | 0.0 |
| VQ-VIB$_\mathcal{C}$ | 32 | 1000 | 1 | 10 | 10 | 0.0 | 0.0 | 0.001 | 0.2 |
| VQ-VIB$_\mathcal{C}$ | 32 | 1000 | 2 | 10 | 10 | 0.0 | 0.0 | 0.001 | 0.4 |
| VQ-VIB$_\mathcal{C}$ | 32 | 1000 | 4 | 10 | 10 | 0.0 | 0.0 | 0.001 | 0.8 |

next paragraph) to classifications for 100 epochs using an Adam optimizer with default parameters, with the learning rate decreasing by a factor of 10 based on plateauing training loss, with a patience of 5 epochs, and early stopping if the learning rate fell below $10^{-8}$.

One particular design choice that we made in finetuning predictors merits elaboration: shifting encodings. Rather than directly training predictors to map from encodings to predictions, we applied a simple linear transformation to the encodings before feeding them to the predictor. Specifically, we multiplied all tensor elements by 5 and increased them by 1. This simple linear transformation does not affect any relations between encodings except scale, and indeed we found that predictors could be successfully trained with this rescaling. Nevertheless, this linear transformation was important to provide a check against merely relying upon initialization conditions for good finetuning performance. In particular, we found that if we did not apply this linear transformation (i.e., pass the raw encodings to the predictor), predictors sometimes performed better than they should given the training data. For example, as a sanity check, we trained a predictor on a binary classification task, but only provided two positive datapoints and no negative data. In general, given this data, one would expect a trained predictor to only predict positive labels. However, we observed that in several cases, the predictor would achieve nearly perfect accuracy, including predicting negative labels for negative inputs. This surprising result disappeared when we simply shifted encodings, indicating that the particular initialization conditions of the predictor seemed to align well with pre-trained encoders. We wanted to measure the effect of data on finetuning performance, rather than just initialization conditions, but we note that this odd phenomenon of well-aligned initializations merits further investigation.

We ran 10 finetuning trials per model, which was important given the small amount of randomly-sampled finetuning data.

## A.3 Hyperparameters

In the following subsections, we present the hyperparameters used for training different encoders in the different domains. In general, we used the following principles when choosing hyperparameters:

- For VQ-based methods, use a large enough codebook to have at least one element per class. Larger $C$ are also acceptable, as tuning weights should decrease the effective codebook size.

- When annealing, use a small enough weight increment to generate smooth changes during training. Larger increments, however, speed up training.

- When annealing for larger $n$ one can increase the annealing rate. Models with greater $n$ tended to use more complex representations, so annealing could be extremely slow for small increments.

### A.3.1 FashionMNIST

For FashionMNIST, we trained all models with batch size 64 for 200 epochs, using the hyperparameters specified in Table 1. The only differences across methods were which hyperparameters we annealed to penalize complexity. Other differences simply reflected differences in architecture (e.g., using a codebook for vector-quantization methods). Pre-training a single model for 200 epochs took approximately 5 minutes on a desktop computer with one NVIDIA 2080 GeForce RTX.

Table 2: Hyperparameters for CIFAR100 training.

| ENCODER | LATENT DIM | $C$ | $n$ | $\lambda_U$ | $\lambda_I$ | $\lambda_{C_0}$ | $\lambda_C$ INCR | $\lambda_{H_0}$ | $\lambda_H$ INCR |
|---|---|---|---|---|---|---|---|---|---|
| $\beta$-VAE | 32 | NA | NA | 10 | 10 | 0.01 | 0.1 | 0.0 | 0.0 |
| VQ-VIB$_\mathcal{N}$ | 32 | 1000 | 1 | 10 | 10 | 0.01 | 0.5 | 0.0 | 0.0 |
| VQ-VIB$_\mathcal{N}$ | 32 | 1000 | 2 | 10 | 10 | 0.01 | 0.5 | 0.0 | 0.0 |
| VQ-VIB$_\mathcal{N}$ | 32 | 1000 | 4 | 10 | 10 | 0.01 | 0.5 | 0.0 | 0.0 |
| VQ-VIB$_\mathcal{C}$ | 32 | 1000 | 1 | 10 | 10 | 0.0 | 0.0 | 0.001 | 0.04 |
| VQ-VIB$_\mathcal{C}$ | 32 | 1000 | 2 | 10 | 10 | 0.0 | 0.0 | 0.001 | 0.08 |
| VQ-VIB$_\mathcal{C}$ | 32 | 1000 | 4 | 10 | 10 | 0.0 | 0.0 | 0.001 | 0.12 |

Table 3: Hyperparameters for iNaturalist training.

| ENCODER | LATENT DIM | $C$ | $n$ | $\lambda_U$ | $\lambda_I$ | $\lambda_{C_0}$ | $\lambda_C$ INCR | $\lambda_{H_0}$ | $\lambda_H$ INCR |
|---|---|---|---|---|---|---|---|---|---|
| $\beta$-VAE | 32 | NA | NA | 10 | 10 | 0.0001 | 0.03 | 0.0 | 0.0 |
| VQ-VIB$_\mathcal{N}$ | 32 | 2000 | 1 | 10 | 10 | 0.001 | 0.1 | 0.0 | 0.0 |
| VQ-VIB$_\mathcal{N}$ | 32 | 2000 | 2 | 10 | 10 | 0.001 | 0.2 | 0.0 | 0.0 |
| VQ-VIB$_\mathcal{N}$ | 32 | 2000 | 4 | 10 | 10 | 0.001 | 0.4 | 0.0 | 0.0 |
| VQ-VIB$_\mathcal{C}$ | 32 | 2000 | 1 | 10 | 10 | 0.0 | 0.0 | 0.001 | 0.05 |
| VQ-VIB$_\mathcal{C}$ | 32 | 2000 | 2 | 10 | 10 | 0.0 | 0.0 | 0.001 | 0.10 |
| VQ-VIB$_\mathcal{C}$ | 32 | 2000 | 4 | 10 | 10 | 0.0 | 0.0 | 0.001 | 0.20 |

### A.3.2 CIFAR100

For CIFAR100, we trained all models with batch size 256 for 400 epochs, using the hyperparameters specified in Table 2. As explained for FashionMNIST, the only substantial differences across architectures were architecture-specific terms that needed to be specified, and which terms were annealed to penalize complexity. Pre-training a single model for 400 epochs took approximately 10 minutes on a desktop computer with one NVIDIA 3080.

### A.3.3 iNaturalist

For iNat, we trained all models with batch size 256, using the hyperparameters specified in Table 3. We trained $\beta$-VAE and VQ-VIB$_\mathcal{C}$ models for 300 epochs, while we trained VQ-VIB$_\mathcal{N}$ models for 600. We used more annealing epochs for VQ-VIB$_\mathcal{N}$ simply because it seemed to need more epochs to eventually anneal to random chance. Likely, a larger annealing rate could also accomplish the desired effect, but initial experiments with faster annealing tended to induce rapid codebook collapse that did not generate the smooth spectrum of MSE values we desired. Pre-training a single model for 300 epochs took approximately 10 minutes on a desktop computer with one NVIDIA 3080 (and twice as long for 600 epochs).

## B  Further Finetuning Results

In the main paper, we included only a small number of graphs highlighting our key results. Here, we include further results that corroborate the main trends stated in the paper. Primarily, these plots include further experiments for varying the amount of finetuning data, as well as varying $n$, the number of codebook elements to combine into a latent representation. We further include details of validation-set experiments, wherein we evaluated a method for autonomously selecting the optimal encoder. Results are divided by domain: FashionMNIST, CIFAR100, and iNat.

### B.1  FashionMNIST

Here, we include the finetuning results for the FashionMNIST domain for varying amounts of finetuning data, ranging over $k \in [1, 2, 5, 10, 50]$, and $n$, the number of quantized vectors to combine into a single representation. Results for each $k$ are included in Figure 10.

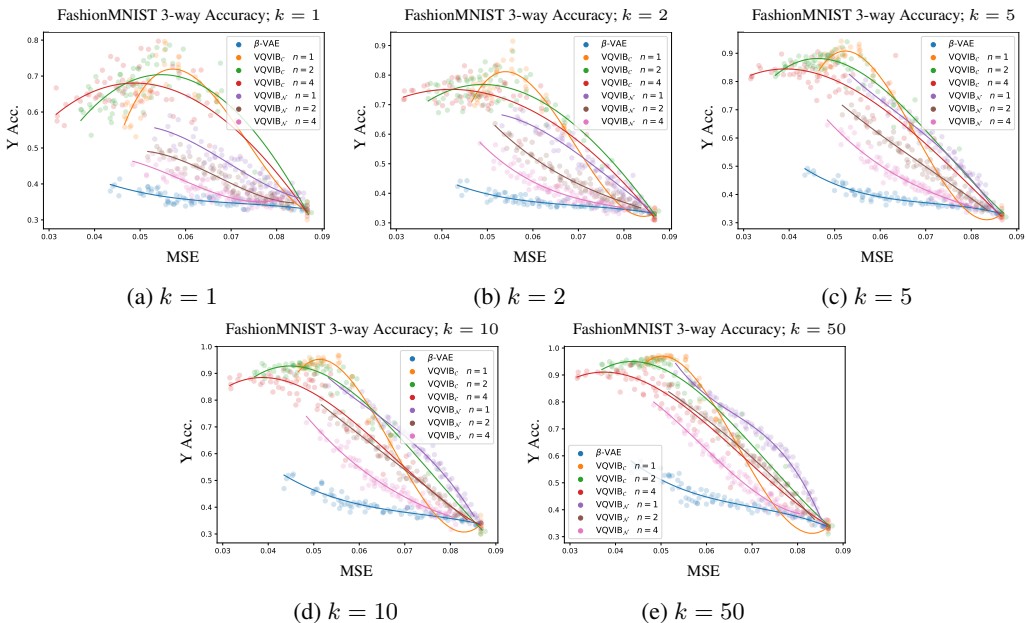

Figure 10: FashionMNIST finetuning results for varying $k$. As $k$ increased, all models benefited. The data-efficiency of advantage of VQ-VIB$_\mathcal{C}$ was most pronounced when using the least amount of data.

Table 4: Finetuning accuracy across domains, using encoders selected via validation set accuracy. For large validation ($v$) and training ($k$) set sizes, this method selected high-performance encoders, but for low $k$ and $v$, it struggled. This motivates our human-in-the-loop framework in the low-data regime.

| | FASHIONMNIST | | | | CIFAR100 2-WAY | | | | INAT 2-WAY | | | |
|---|---|---|---|---|---|---|---|---|---|---|---|---|
| $v$ | $k=2$ | 5 | 10 | 50 | 2 | 5 | 10 | 50 | 2 | 5 | 10 | 50 |
| 1 | 0.73 | 0.89 | 0.93 | 0.97 | 0.71 | 0.79 | 0.83 | 0.90 | 0.65 | 0.79 | 0.89 | 0.90 |
| 5 | – | – | 0.94 | 0.97 | – | – | 0.82 | 0.91 | – | – | 0.88 | 0.90 |
| 10 | – | – | – | 0.97 | – | – | – | 0.91 | – | – | – | 0.89 |

As expected, increasing the amount of finetuning data improved performance for all models, and the gap between all model types (VQ-VIB$_\mathcal{C}$, VQ-VIB$_\mathcal{N}$, and $\beta$-VAE) shrank. It is noteworthy, however, that a VQ-VIB$_\mathcal{C}$ model, tuned to the right complexity level and trained with just one example per ternary class (Figure 10 a), achieved better accuracy than a $\beta$-VAE model trained with 50 examples per class (Figure 10 e). Further, for any fixed $k$, VQ-VIB$_\mathcal{C}$ consistently outperformed VQ-VIB$_\mathcal{N}$, suggesting that many recent works that use VQ-VIB$_\mathcal{N}$ could be improved by replacing the model type [27, 28, 12, 7]. Lastly, for both VQ-VIB$_\mathcal{N}$ and VQ-VIB$_\mathcal{C}$, increasing $n$ tended to support lower MSE but worse finetuning accuracy. This supports an intuition that combining more discrete representations starts to more densely fill the representation space, trending towards continuous representations.

We note briefly that VQ-VIB$_\mathcal{N}$, both in this domain and others (explored in the next sections), typically failed to learn as complex representations as either VQ-VIB$_\mathcal{C}$ or $\beta$-VAEs. This is apparent given the limited range of MSE values for the VQ-VIB$_\mathcal{N}$ curves. We consistently struggled to make VQ-VIB$_\mathcal{N}$ learn as rich representations as for the other model types, which led to worse reconstructions and higher MSE values.

Lastly, Table 4 includes results from our validation experiments for all three experiment domains. Recall that we tested a method for autonomously selecting the best encoder, among the suite of encoders of different complexity levels, by measuring validation set accuracy. Table 4 includes results from such experiments for different validation set sizes ($v$) and different $k$. For a given $k$ and $v$, we randomly sampled $v$ datapoints per class label to be part of the validation set and used the remaining

data for finetuning. For large $k$, this method worked quite well by selecting high-accuracy encoders. However, for small $k$ and $v$, validation set accuracy was a noisy proxy for model performance (because of the small validation set size), so performance tended to be suboptimal. For example, in the FashionMNIST domain, for $k = 2, v = 1$, models achieved mean performance of 73%, lower than the over than 80% accuracy achieved by tuning to the right complexity (Figure 10 b). Overall, these validation set experiments confirm that, for large enough $k$, one may autonomously select optimal encoders, but for very small $k$, such autonomous methods fail.

## B.2 CIFAR100

We found similar trends in the CIFAR100 to those in the FashionMNIST domain and plotted results in Figures 11 and 12 (for 2-way and 20-way finetuning tasks, respectively). In all experiments, VQ-VIB$_\mathcal{C}$ outperformed both $\beta$-VAE and VQ-VIB$_\mathcal{N}$. In the 2-way finetuning example, we again found a peaked curve for VQ-VIB$_\mathcal{C}$ finetuning accuracy as a function of MSE, indicating that tuning to the right complexity level induced the best accuracy. In the more complex 20-way classification task, however, we did not observe this peak.

This last result is unsurprising: the 20-way hierarchy in CIFAR100 is less semantically meaningful and likely less obvious in photos than the 2-way task of distinguishing living and non-living things. For example, two of the 20 categories are simply different sorts of vehicles. It would be extremely surprising for VQ-VIB$_\mathcal{C}$ to learn such arbitrary groups automatically while compressing representations. Without learning the right groupings, VQ-VIB$_\mathcal{C}$ cannot benefit from learning less complex representations.

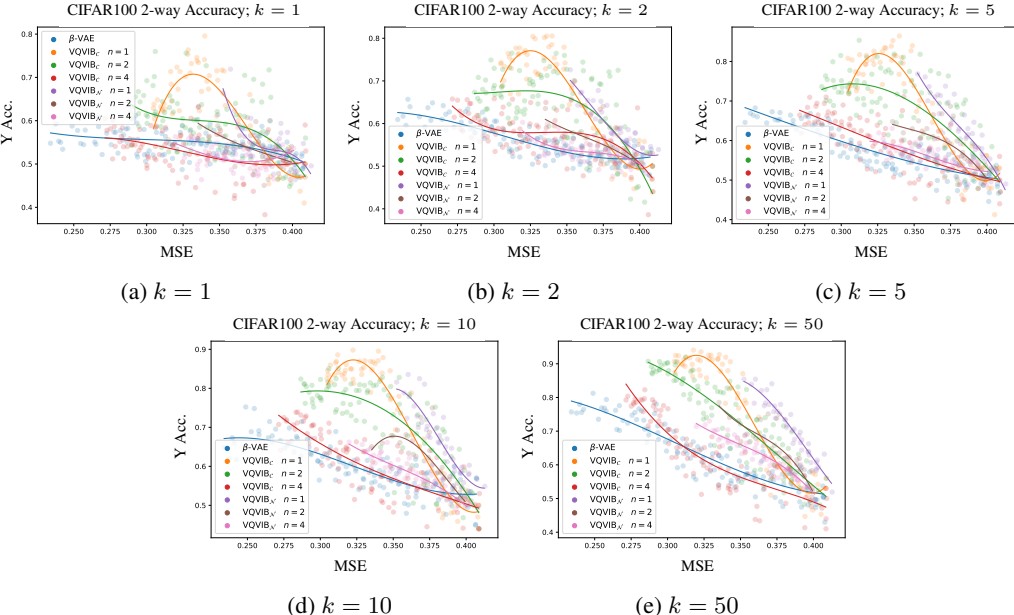

Figure 11: CIFAR100 2-way finetuning results for varying $k$.

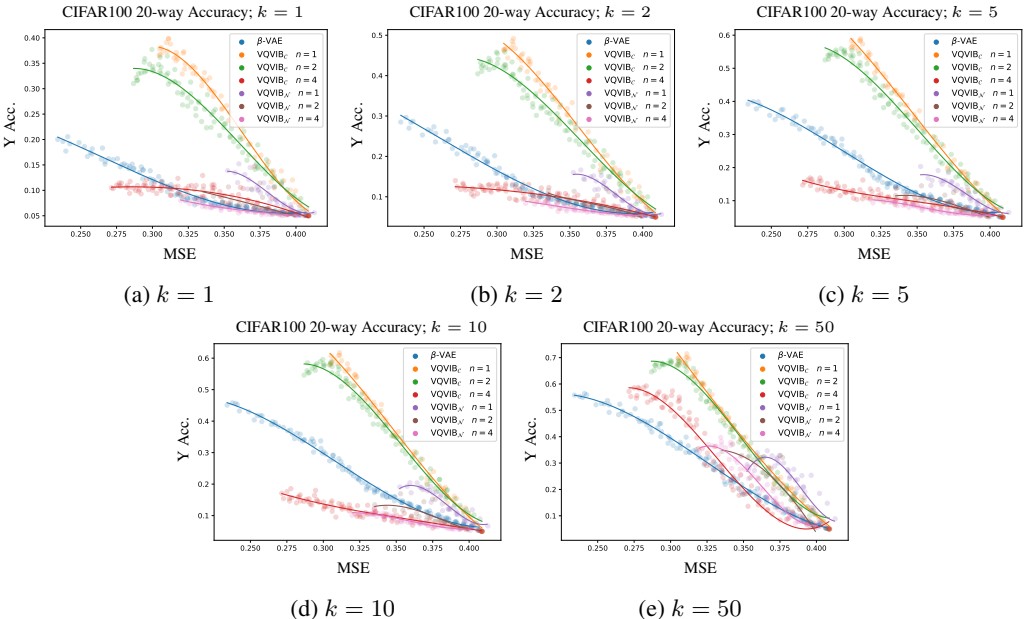

Figure 12: CIFAR100 20-way finetuning results for varying $k$.

### B.3 iNaturalist

Lastly, we found similar trends in finetuning in the iNat domain, finetuned on a 3-way (Figure 13), 34-way (Figure 14), and 1010-way (Figure 15) finetuning task.

On the 3-way finetuning task (between animals, plants, and fungi), we observed similar peaking behavior as in earlier experiments, indicating yet again the importance of tuning to the right complexity. In addition, as in prior results, we found a similar trend that greater $n$ tended to allow greater complexity (lower MSE) but induced worse finetuning performance. For example, in Figure 13 b, the orange line, corresponding to $n = 1$ stays above and to the right of the green ($n = 2$) and red ($n = 4$) lines. Intuitively, this seems to indicate that the more combinatorial representations, with greater $n$, were somewhat of a midpoint between the continuous $\beta$-VAE representations and the discrete representations used by VQ-VIB$_\mathcal{C}$ for $n = 1$.

Results from the 34-way finetuning followed similar patterns as before as well. Just as in CIFAR100 wherein we tested both a 2-way and 20-way finetuning task, this 34-way finetuning task for iNat showed that VQ-VIB$_\mathcal{C}$ continued to outperform VQ-VIB$_\mathcal{N}$ and $\beta$-VAE for more complex finetuning tasks, although the performance gap shrank as $k$ increased.

Most interestingly, perhaps, we conducted yet another iNat finetuning experiment, this time using the 1010 low-level labels that had originally been used during pre-training. As before, we used very small amounts of data in finetuning (e.g., for $k = 1$, only 1 example from each class, so 1010 labeled examples total). Results from those experiments are shown in Figure 15.

For small $k$, we again see that VQ-VIB$_\mathcal{C}$ outperforms other model types. For larger $k$, however, we see one of the limitations of VQ-VIB$_\mathcal{C}$. Because the discrete encoders learned less complex representations than $\beta$-VAEs (as shown by the fact that they never reach lower MSE values), with enough finetuning data, $\beta$-VAEs are able to capture distinctions between classes that VQ-VIB$_\mathcal{C}$ models cannot. Thus, in the particular case of large amounts of finetuning data and complex finetuning tasks, more complex, continuous encoders continue to outperform our method.

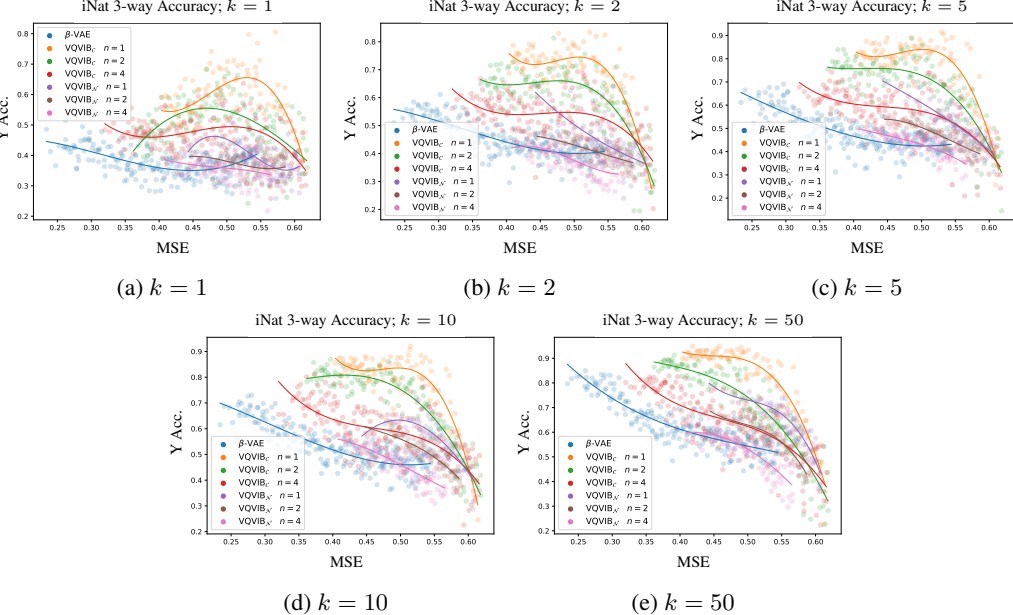

Figure 13: iNat 3-way finetuning results for varying $k$.

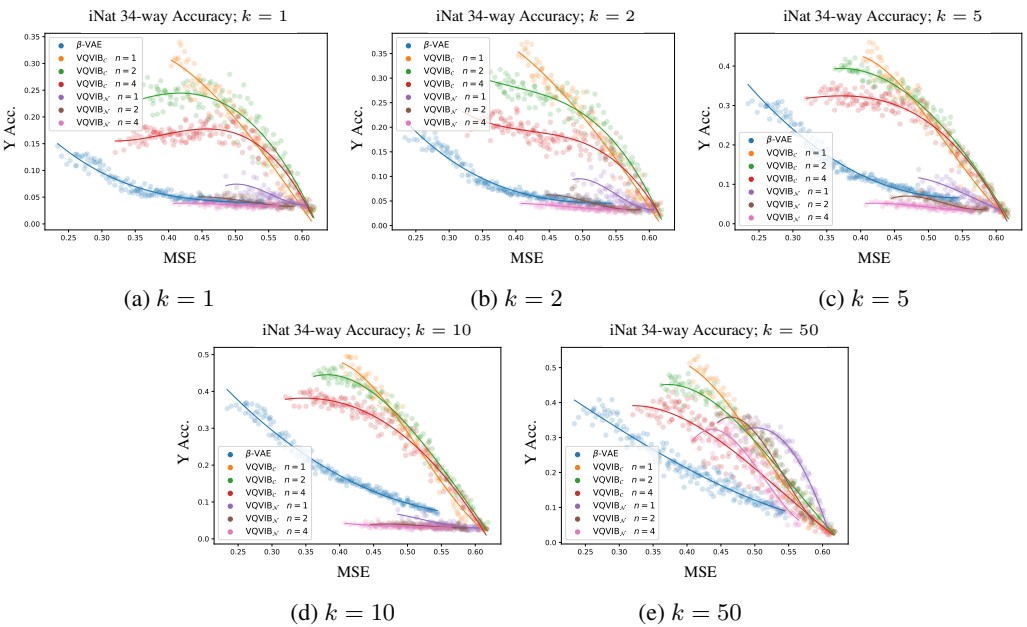

Figure 14: iNat 34-way finetuning results for varying $k$.

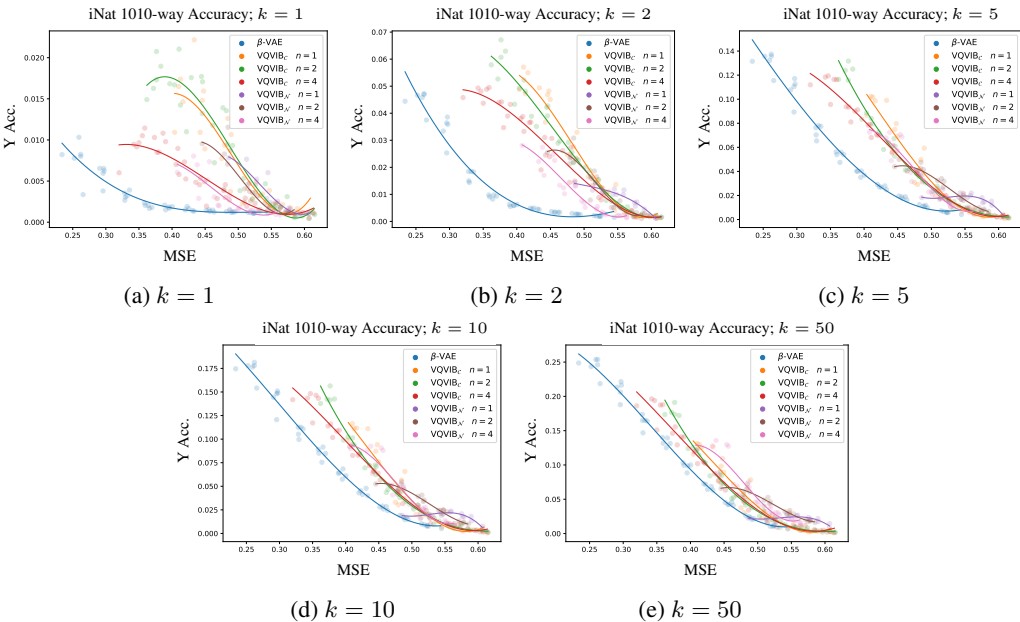

Figure 15: iNat 1010-way finetuning results for varying $k$.

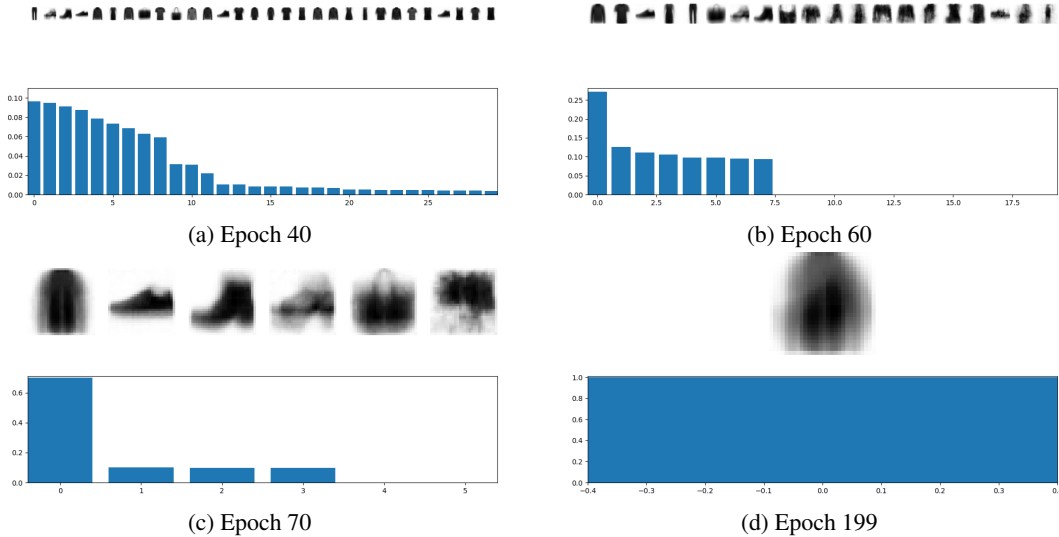

(a) Epoch 40

(b) Epoch 60

(c) Epoch 70

(d) Epoch 199

Figure 16: The evolution of the distribution over prototypes during annealing for VQ-VIB$_\mathcal{C}$ in the FashionMNIST domain. In early epochs, VQ-VIB$_\mathcal{C}$ uses many prototypes, with a long-tailed distribution. Over the course of annealing the entropy, the probability distribution becomes more concentrated (b and c) before collapsing to a single prototype (d).

## C  Prototype Utilization: Further Visualizations

Here, we include some further visualizations that we omitted from the main paper due to space constraints. These visualization primarily illustrate the importance of entropy-regulated representation learning (for VQ-VIB$_\mathcal{C}$) vs. complexity-regulated (for VQ-VIB$_\mathcal{N}$).

Figures 16 and 17 shows the prototypes for VQ-VIB$_\mathcal{C}$ and VQ-VIB$_\mathcal{N}$, respectively, in the FashionMNIST domain over the course of training. Each subfigure consists of a top row of decoded prototypes, with associated probabilities (frequency of use measured when passing through images from the test set) below. The 30 most frequent prototypes are visualized, or fewer prototypes if fewer were used.

There is an important trend in Figures 16 and 17: the entropy-based annealing for VQ-VIB$_\mathcal{C}$ caused models to use fewer prototypes, while the complexity-based annealing for VQ-VIB$_\mathcal{N}$ did not. At epoch 40, just as both methods begin annealing, VQ-VIB$_\mathcal{C}$ and VQ-VIB$_\mathcal{N}$ use a large number of prototypes, as seen by the long-tailed distributions. Over the course of annealing, however, VQ-VIB$_\mathcal{C}$ uses fewer prototypes, and merges images of different classes into the same prototype. Thus, the degenerate encoder at the end of annealing (epoch 199) uses just a single prototype to represent all possible inputs (Figure 16). At the same time, VQ-VIB$_\mathcal{N}$, during annealing, does not use fewer prototypes. Rather, the complexity-penalization term seems to induce the model to make the mapping from input to prototype more stochastic (Figure 17). Thus, the degenerate VQ-VIB$_\mathcal{N}$ encoder uses many prototypes, each of which is blurry because it could correspond to any input.

Visualizations of decoded prototypes for the CIFAR100 domain is more challenging. In richer image domains, prototype-based methods often use training examples as prototypes [3, 18, 4], which can make it more difficult to understand when a single prototype represents more than one concept. Nevertheless, by visualizing the distribution over prototypes (without decoding them), we see the same pattern that VQ-VIB$_\mathcal{C}$ tends to learn to use fewer prototypes over the course of annealing than VQ-VIB$_\mathcal{N}$. Snapshots of the categorical distributions for CIFAR100 are included in Figure 18.

## D  Ablation Study: Entropy vs. Complexity

Here, we present results motivating penalizing entropy, as opposed to complexity, in VQ-VIB$_\mathcal{C}$. Appendix C showed how annealing entropy in VQ-VIB$_\mathcal{C}$ caused models to use fewer prototypes, whereas penalizing complexity in VQ-VIB$_\mathcal{N}$ did not induce similar reductions in effective codebook

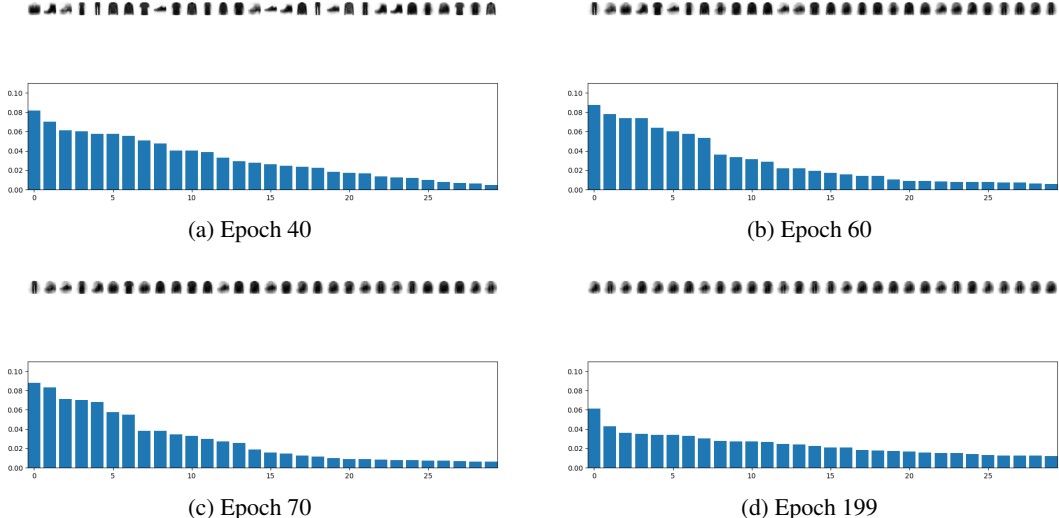

Figure 17: The evolution of the distribution over prototypes during annealing for VQ-VIB$_\mathcal{N}$ in the FashionMNIST domain. Unlike annealing entropy for VQ-VIB$_\mathcal{C}$, annealing the complexity in VQ-VIB$_\mathcal{N}$ did not result in fewer prototypes being used. Instead, each prototype became blurrier, indicating that each prototype became more likely regardless of class.

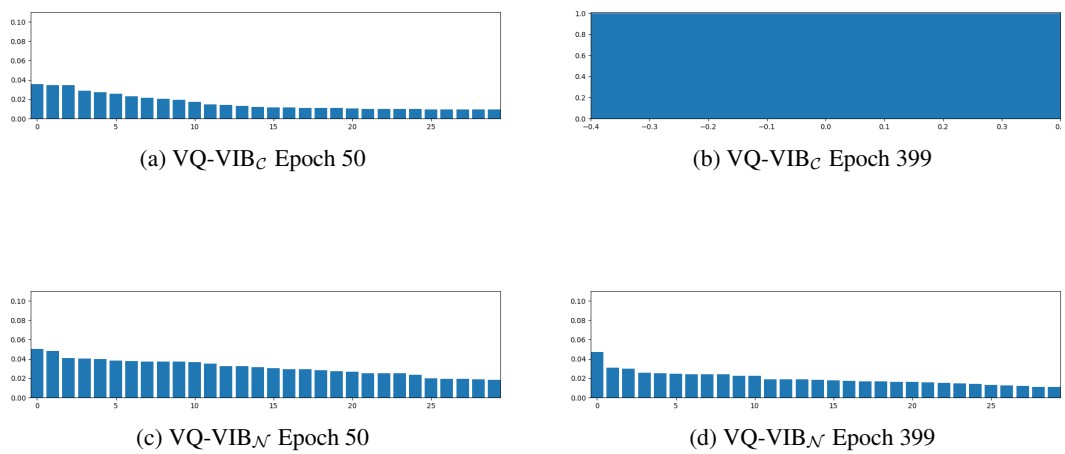

Figure 18: Categorical distribution over prototypes while annealing in the CIFAR100 domain at the start of annealing (Epoch 50) and at the end (Epoch 399) for VQ-VIB$_\mathcal{C}$ (top row) and VQ-VIB$_\mathcal{N}$ (bottom row). Annealing the entropy in VQ-VIB$_\mathcal{C}$ caused the model to use fewer prototypes (note the degenerate categorical distribution over only one prototype at Epoch 399), whereas annealing complexity for VQ-VIB$_\mathcal{N}$ did not cause similar concentration.

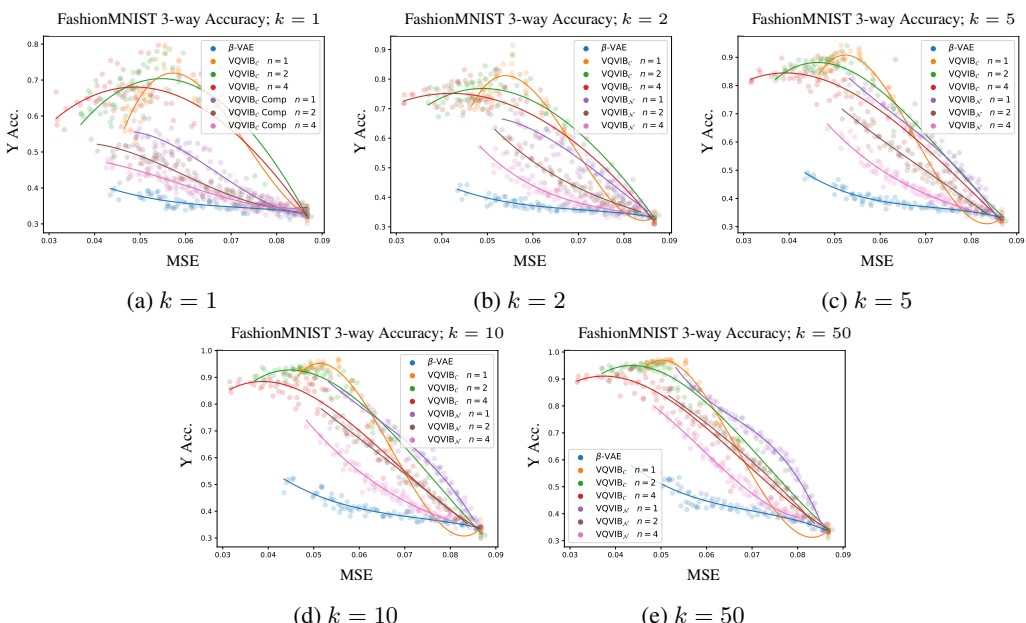

(a) $k = 1$       (b) $k = 2$       (c) $k = 5$

(d) $k = 10$       (e) $k = 50$

Figure 19: FashionMNIST finetuning results for varying $k$, comparing annealing by entropy (VQ-VIB$_{\mathcal{C}}$) and annealing by complexity (VQ-VIB$_{\mathcal{C}}$ Comp.). Annealing by complexity resulted in worse finetuning performance.

size. Further experiments corroborate our findings that penalizing entropy was the key to this difference in behavior.

We trained VQ-VIB$_{\mathcal{C}}$ agents on the FashionMNIST task, using the same pre-training and finetuning procedures as in the main paper, with the only difference being that we annealed the complexity of representations instead of the entropy. Results from finetuning such models are included in Figure 19.

Figure 19 shows that annealing by entropy, as opposed to complexity, was the key factor in improving VQ-VIB$_{\mathcal{C}}$ finetuning performance. The difference in performance when penalizing entropy vs. complexity closely matches the difference in performance between VQ-VIB$_{\mathcal{C}}$ and VQ-VIB$_{\mathcal{N}}$ examined in the main paper. Thus, the entropy-regularization term seems to explain much of the difference between VQ-VIB$_{\mathcal{C}}$ and VQ-VIB$_{\mathcal{N}}$.

In subsequent experiments in the CIFAR100 and iNat domains, therefore, we tested whether penalizing the estimated entropy of VQ-VIB$_{\mathcal{N}}$ models matched VQ-VIB$_{\mathcal{C}}$ results from the main paper. We note that Tucker et al. [27] advocate for a small positive $\lambda_H$ to penalize entropy, but the authors also acknowledge that exactly computing this entropy is impossible given the VQ-VIB$_{\mathcal{N}}$ architecture.

Finetuning results for CIFAR100 (on the 2-way and 20-way finetuning tasks) and iNat (on the 3-way and 34-way finetuning tasks), for VQ-VIB$_{\mathcal{C}}$ trained by varying $\lambda_C$ and VQ-VIB$_{\mathcal{N}}$ trained by varying $\lambda_H$, are included in Figures 20, 21, 22, and 23. Several important trends emerge from viewing these plots, especially compared to results from our main paper for VQ-VIB$_{\mathcal{C}}$ controlled via $\lambda_H$.

First, finetuning performance is noisier using these models compared to results from the main text. This likely arises, for VQ-VIB$_{\mathcal{N}}$ models, because increasing $\lambda_H$ failed to consistently reduce the number of discrete representations used. Thus, for a given MSE value, different models used different numbers of representations, and therefore exhibited different finetuning performance.

Second, varying $\lambda_H$, instead of $\lambda_C$, seemed to somewhat improve VQ-VIB$_{\mathcal{N}}$ performance, but not as much as when varying $\lambda_H$ for VQ-VIB$_{\mathcal{C}}$, as presented in our main paper. For example, consider Figure 21 a. The best-performing model, VQ-VIB$_{\mathcal{N}}$, $n = 1$, peaks at finetuning accuracy of approximately 0.16, outperforming VQ-VIB$_{\mathcal{C}}$ models when varying $\lambda_C$. However, in Figure 12 a, we found that VQ-VIB$_{\mathcal{C}}$ models in the exact same setting achieved a mean accuracy of approximately 0.38: more than double the VQ-VIB$_{\mathcal{N}}$ performance. Thus, varying $\lambda_H$ seemed to improve VQ-VIB$_{\mathcal{N}}$

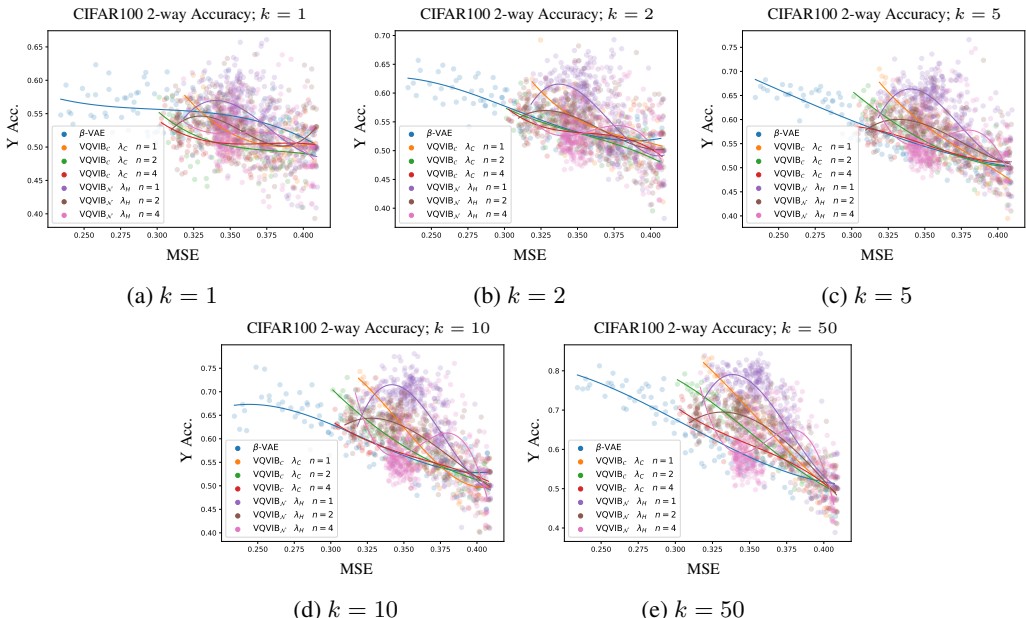

Figure 20: Ablation study results for the CIFAR100 2-way finetuning task. Tuning $\lambda_C$ for VQ-VIB$_\mathcal{C}$ or $\lambda_H$ for VQ-VIB$_\mathcal{N}$ led to worse results than in our main paper.

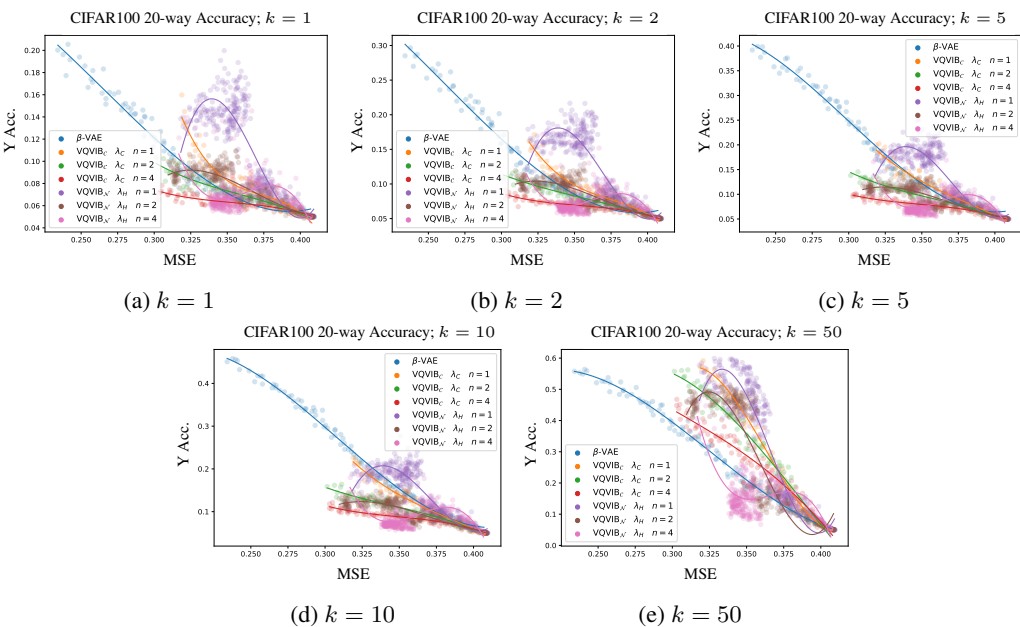

Figure 21: Ablation study results for the CIFAR100 20-way finetuning task. Tuning $\lambda_C$ for VQ-VIB$_\mathcal{C}$ or $\lambda_H$ for VQ-VIB$_\mathcal{N}$ led to worse results than in our main paper.

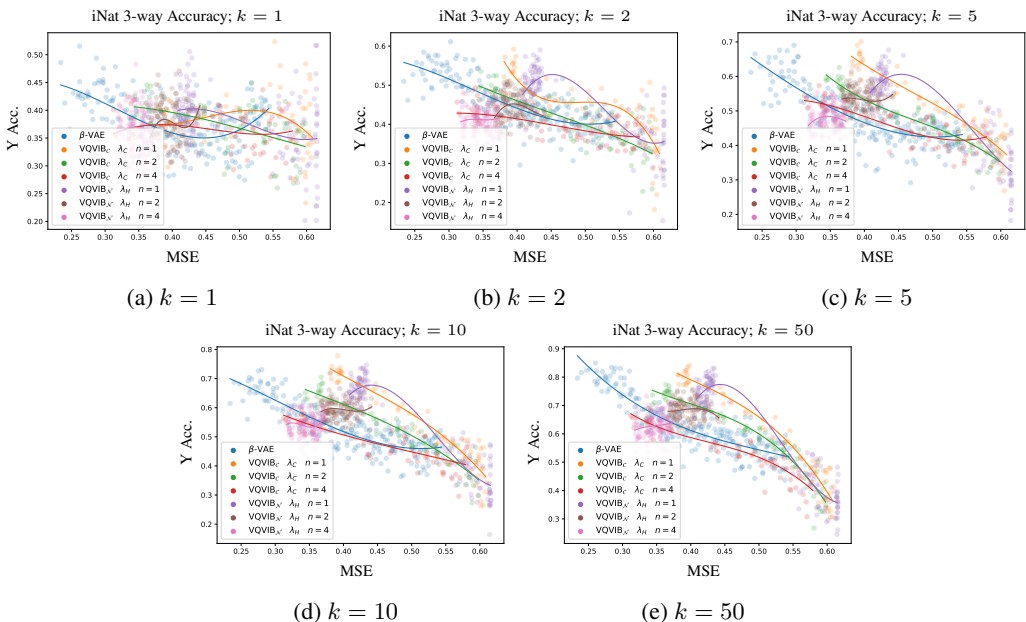

Figure 22: Ablation study results for the iNat 3-way finetuning task. Tuning $\lambda_C$ for VQ-VIB$_{\mathcal{C}}$ or $\lambda_H$ for VQ-VIB$_{\mathcal{N}}$ led to worse results than in our main paper.

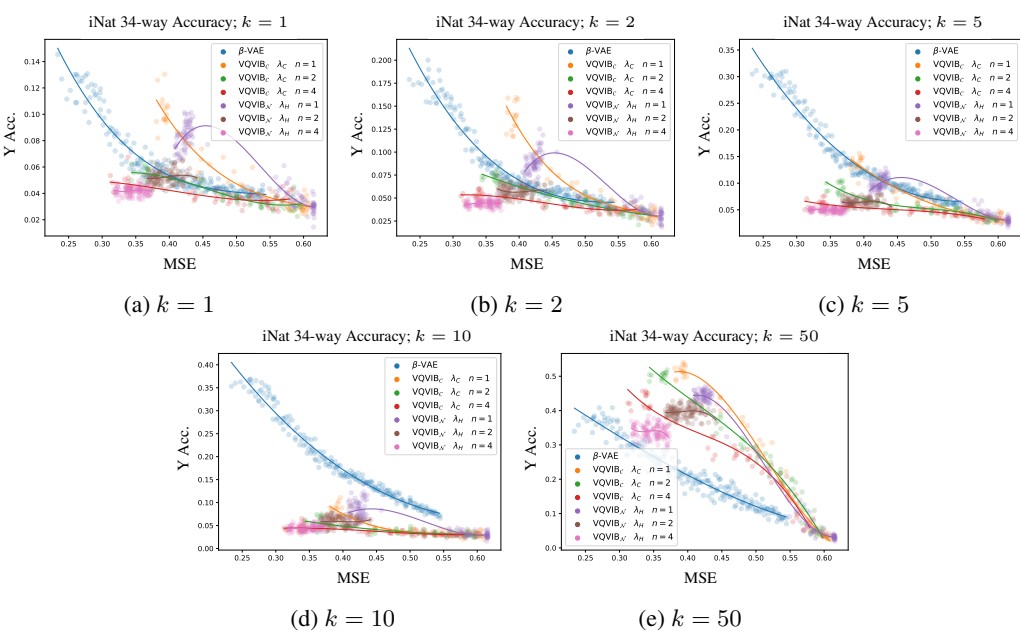

Figure 23: Ablation study results for the iNat 34-way finetuning task. Tuning $\lambda_C$ for VQ-VIB$_{\mathcal{C}}$ or $\lambda_H$ for VQ-VIB$_{\mathcal{N}}$ led to worse results than in our main paper.

performance somewhat, but VQ-VIB$_\mathcal{C}$ better supports penalizing entropy, and therefore achieves higher performance.

Third, varying $\lambda_C$, instead of $\lambda_H$, for VQ-VIB$_\mathcal{C}$ worsened finetuning performance. Once again, by comparing finetuning performance for VQ-VIB$_\mathcal{C}$ models in Figure 21 a (achieving a maximum accuracy around 0.14), to results from our main paper, we note the importance of penalizing the entropy of representations.

Thus, in general these ablation studies support many of the design decisions made in the main paper.

1. Varying $\lambda_H$, instead of $\lambda_C$ for VQ-VIB$_\mathcal{C}$ improves finetuning performance by decreasing the number of discrete representations used.
2. VQ-VIB$_\mathcal{N}$ benefits somewhat from penalizing entropy, but because it is architecturally unable to support exact calculations of entropy, we were unable to match VQ-VIB$_\mathcal{N}$ performance.

It is certainly possible that some optimal combination of $\lambda_H$ and $\lambda_C$ might further improve VQ-VIB$_\mathcal{N}$ or VQ-VIB$_\mathcal{C}$ performance; initial explorations of such combinations with fixed $\lambda_H$ values while annealing $\lambda_C$ did not yield obvious results. Most importantly, our current findings are enough to indicate that controlling the entropy of discrete representations appears important for data-efficient finetuning.

# E   User Study Results

Here, we include complete results from our user study. Table 5 includes accuracy rates for all model types and visualizations, for all three questions. For Questions 2 and 3, selecting the lowest-MSE (highest complexity) encoder was correct, so accuracy rates for all model and visualization types for these questions remained high. For Question 1, however, selecting the correct encoder required users to select non-maximally-complex representations; for this question, only users viewing prototypes of VQ-VIB$_\mathcal{C}$ models performed above random chance.

Table 5: Accuracy rates by encoder type and visualization method for the three questions (reporting means and standard errors over 20 subjects). For the most challenging question (Question 1), visualizing VQ-VIB$_\mathcal{C}$ prototypes supported the greatest accuracy.

| Model | Viz. | Question 1 | Question 2 | Question 3 |
|-------|------|-----------|-----------|-----------|
| VQ-VIB$_\mathcal{C}$ | MSE | 0.10 (0.02) | 0.70 (0.05) | 0.85 (0.03) |
| VQ-VIB$_\mathcal{C}$ | Proto | 0.55 (0.06) | 0.75 (0.04) | 0.90 (0.02) |
| VQ-VIB$_\mathcal{N}$ | MSE | 0.10 (0.02) | 0.80 (0.04) | 0.85 (0.03) |
| VQ-VIB$_\mathcal{N}$ | Proto | 0.10 (0.02) | 0.90 (0.02) | 0.90 (0.02) |

We further included full results for our Mixed Linear Effects Modeling statistical tests in Table 6. All but two effects are not significant at the $p < 0.05$ level. The two significant effects are 1) Question 1 had a significant *negative* effect on accuracy rate, and 2) there was a significant *positive* interaction effect between visualizing VQ-VIB$_\mathcal{C}$ prototypes and Question 1. Jointly, these two effects show that Question 1 was harder for users than the other questions, but that seeing VQ-VIB$_\mathcal{C}$ prototypes to some extent mitigated this increased difficulty.

Table 6: Results of the Mixed Linear Effects Modeling of user responses, predicting accuracy as a function of model and visualization type and Question. Our key results is that there is a significant positive interaction effect between visualizing VQ-VIB$_\mathcal{C}$ prototypes and performing better on Question 1 (bolded for emphasis). The intercept row corresponds to Question 2 with VQ-VIB$_\mathcal{C}$-MSE.

| | Coef. | Std.Err. | z | P$|z|$ | [0.025 | 0.975] |
|---|-------|----------|---|--------|--------|--------|
| Intercept | 0.700 | 0.084 | 8.356 | 0.000 | 0.536 | 0.864 |
| VQ-VIB$_\mathcal{C}$-Proto | 0.050 | 0.118 | 0.421 | 0.674 | -0.182 | 0.282 |
| VQ-VIB$_\mathcal{N}$-MSE | 0.100 | 0.118 | 0.845 | 0.398 | -0.132 | 0.332 |
| VQ-VIB$_\mathcal{N}$-Proto | 0.200 | 0.119 | 1.692 | 0.091 | -0.032 | 0.433 |
| Question 1 | -0.600 | 0.118 | -5.085 | 0.000 | -0.831 | -0.369 |
| Question 3 | 0.150 | 0.118 | 1.271 | 0.204 | -0.081 | 0.381 |
| **VQ-VIB$_\mathcal{C}$-Proto:Question 1** | **0.400** | **0.167** | **2.397** | **0.017** | **0.073** | **0.727** |
| VQ-VIB$_\mathcal{N}$-MSE:Question 1 | -0.100 | 0.167 | -0.599 | 0.549 | -0.427 | 0.227 |
| VQ-VIB$_\mathcal{N}$-Proto:Question 1 | -0.200 | 0.167 | -1.199 | 0.231 | -0.527 | 0.127 |
| VQ-VIB$_\mathcal{C}$-Proto:Question 3 | -0.000 | 0.167 | -0.000 | 1.000 | -0.327 | 0.327 |
| VQ-VIB$_\mathcal{N}$-MSE:Question 3 | -0.100 | 0.167 | -0.599 | 0.549 | -0.427 | 0.227 |
| VQ-VIB$_\mathcal{N}$-Proto:Question 3 | -0.150 | 0.167 | -0.899 | 0.369 | -0.477 | 0.177 |
| Group Var | 0.001 | 0.024 | | | | |

# F   User Study

In the subsequent pages, we have included the exact pdf document of a survey shared with participants of the user study. This pdf was generated for a participant viewing VQ-VIB$_\mathcal{C}$ prototypes; similar surveys were populated with data for other models (VQ-VIB$_\mathcal{N}$) or visualization methods (MSE plots). Furthermore, the order of the three questions was randomized to avoid ordering effects.

**Block 1**

# Brief

***Please do not take this study on a mobile phone, the text and images will not display correctly.***

You are free to leave this study at any time.

We will (1) explain the survey format, (2) include two examples of how to complete the survey, and then (3) ask three questions. In total, the survey will take around five minutes.

After you are finished with the survey, you will be redirected to Prolific to recieve your payment.

***Please do not take this study if you have done a similar one before.***

Thank you for your participation!

**Block 11**

Please enter your Prolific ID

| |
|---|

**Block 2**

# Introduction

A computer program is good at classifying clothes into these 10 categories

- T-shirt
- Trouser

- Pullover
- Dress
- Coat
- Sandal
- Shirt
- Sneaker
- Bag
- Ankleboot

See below for example visualizations of these 10 categories

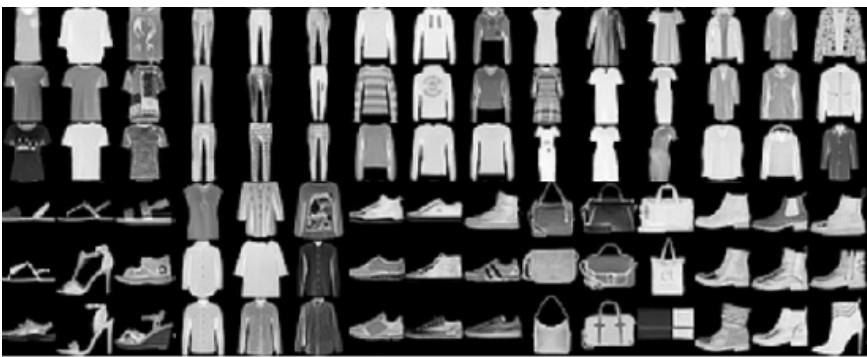

**Block 3**

# Introduction

However, imagine that you don't care about these 10 specific categories.

Instead, you want to categorize them into more high-level groups.

For example, you may only want to sort your clothes into these more general 3 groups:

- **Group 1**: t-shirt, shirt, pullover, and dress
- **Group 2:** boot, sandal, ankleboot, and sneaker
- **Group 3**: bag and trouser

**Block 10**

Your task is the following:

**Step 1:** Read a description of the categories that the computer program needs to sort items of clothing into. These categories will change for different questions, **so make sure to re-read the categories**.

**Step 2:** Look at visualizations of different computer programs.

**Step 3:** Based on the visualizations from Step 2, select which of the programs you think will be best able to accomplish the task described in **Step 1.** In general, try to select the visualization that 1) can represent the different categories we ask for and 2) has the fewest representations, while still having enough representations to distinguish between categories.

On the next page, we have included an example of how to follow these steps.

**Example Question**

# Example Question 1:

**Step 1:**
The categories are:

1. Group 1:  T-shirt, Shirt, Pullover
2. Group 2:  Purse

**Step 2:**
Select which of the following visualizations best reflects the categories from Step 1.

Option 1

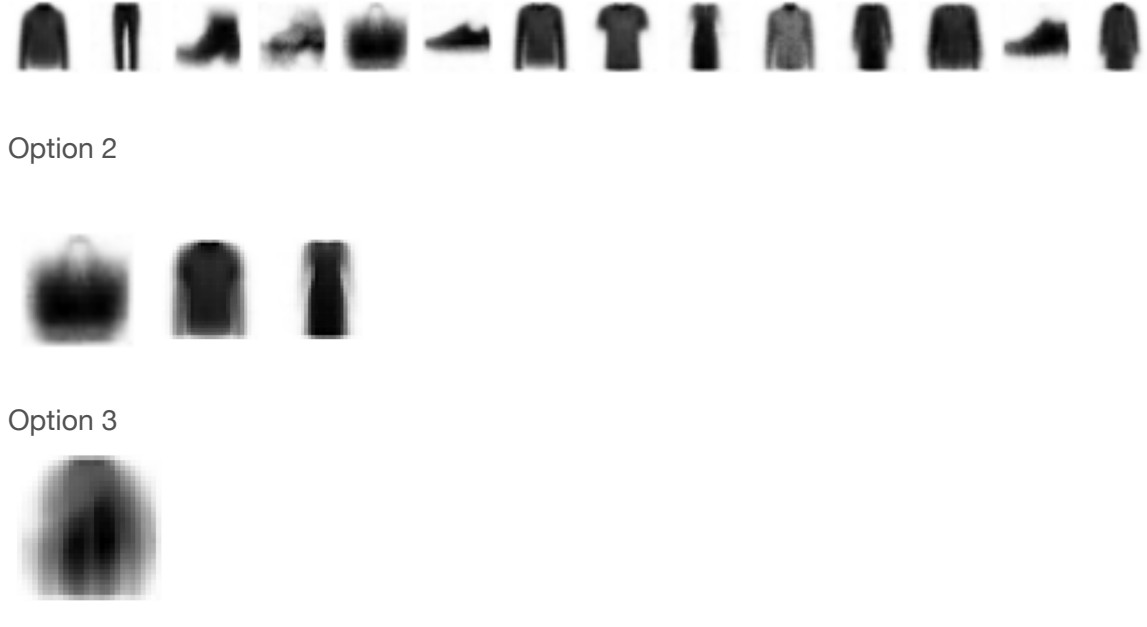

Option 2

Option 3

**Option 2 is best for this task** because it demonstrates visualizations of the two groupings that we care about here (distinguishing between purses and different types of shirts) while using few representations**.**

**Options 1** has enough representations for the two groups we care about, but it has more representations than Option 2, so Option 2 is a better choice**.**

**Option 3 is too simple** and therefore cannot tell the two groupings apart**.**

**Step 3 (Example 1):**
Please select which option you think best represents the groupings listed:

- ○ Option 1
- ○ Option 2
- ○ Option 3

## Example Question 2:

**Step 1:**
The categories are:

1. Group 1:  T-shirt
2. Group 2:  Pullover and Shirt
3. Group 3:  Purse

**Step 2:**
Select which of the following visualizations best reflects the categories from Step 1.

Option 1

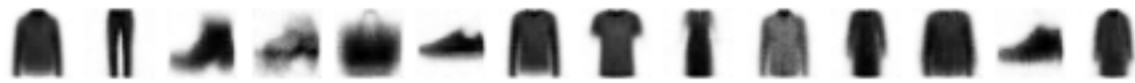

Option 2



Option 3

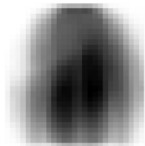

**Option 1 is best for this task** because it demonstrates visualizations of the three groupings that we care about here. Because we care about distinguishing between t-shirts and shirts (Group 1 vs. 2), we need to be able to tell t-shirts apart based on the visualizations.

**Options 2 and 3 are bad choices** because they cannot tell t-shirts (Group 1) apart from other pullovers or shirts (Group 2). Therefore, even though they use fewer representations

**Step 3 (Example 2):**

Please select which option you think best represents the groupings listed:

○ Option 1
○ Option 2
○ Option 3

**Block 10**

# Please only participate in this study if you are confident you understand the instructions correctly

**Block 11**

# Click next to begin the study

**Question 1**

## Question:

**Step 1:**

The computer program needs to sort the items into the following 3 categories:

- 1. Group 1: T-shirt, Trouser, Sandal
- 2. Group 2: Pullover, Dress, Sneaker
- 3. Group 3: Coat, Shirt, Bag, Ankle Boot

**Step 2:**

Please look at the options below:

Option 1

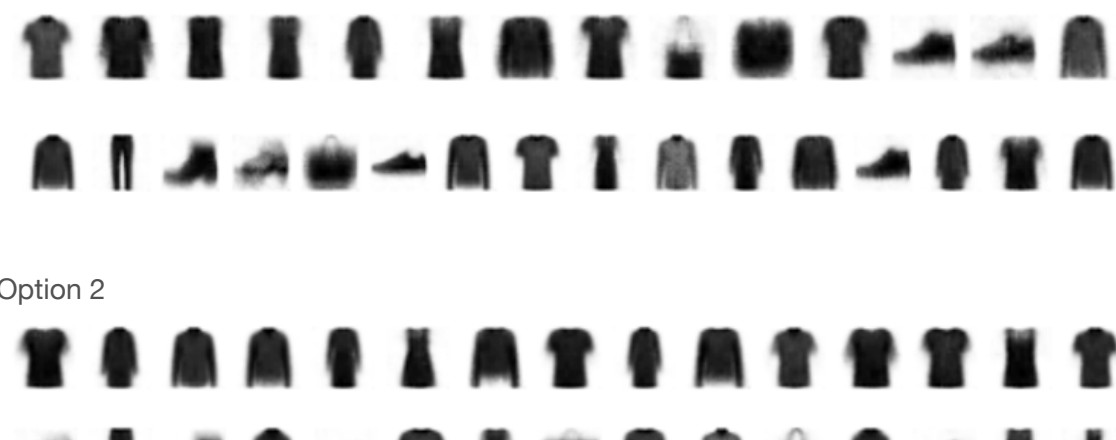

Option 2

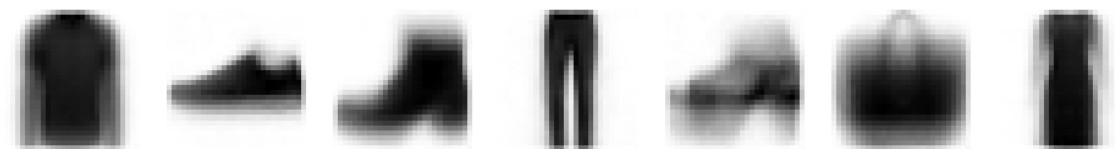

Option 3

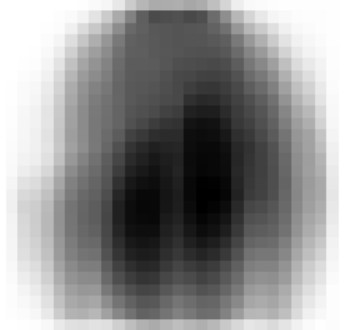

Option 4

**Step 3:**

Please select which option you think best represents the groupings listed:

- ○ Option 1
- ○ Option 2
- ○ Option 3
- ○ Option 4

**Question 2**

# Question:

**Step 1:**

The computer program needs to sort the items into the following 3 categories:

- 1. Group 1: T-shirt, Pullover, Coat, Shirt
- 2. Group 2: Trouser, Dress, Bag
- 3. Group 3: Sandal, Sneaker, Ankle boot

**Step 2:**

Please look at the options below:

Option 1

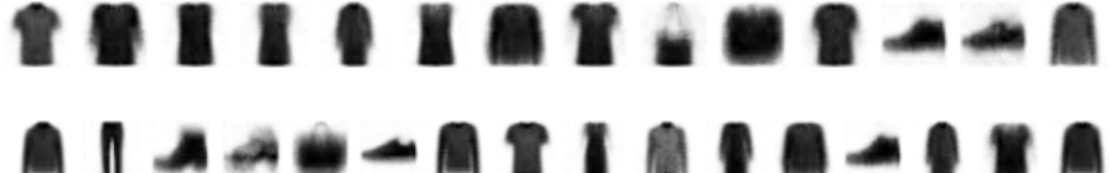

Option 2

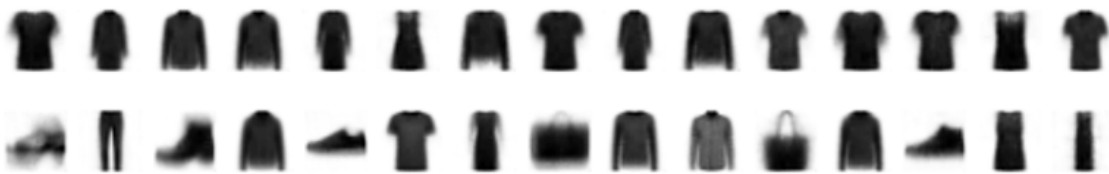

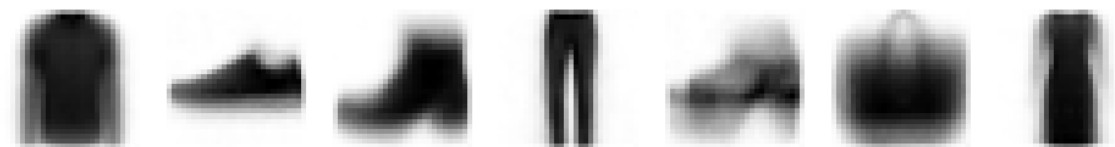

Option 3

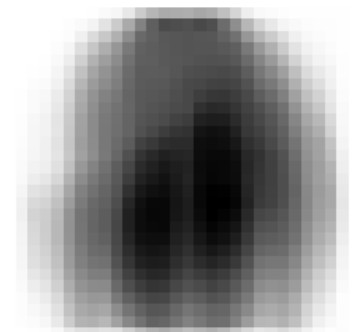

Option 4

**Step 3:**

Please select which option you think best represents the groupings listed:

- ○ Option 1
- ○ Option 2
- ○ Option 3
- ○ Option 4

**Question 3**

# Question:

**Step 1:**

The computer program needs to sort the items into the following 10 categories:

- 1. T-shirt
- 2. Trouser
- 3. Pullover
- 4. Dress
- 5. Coat
- 6. Sandal
- 7. Shirt
- 8. Sneaker
- 9. Bag
- 10. Ankleboot

**Step 2:**

Please look at the options below:

Option 1

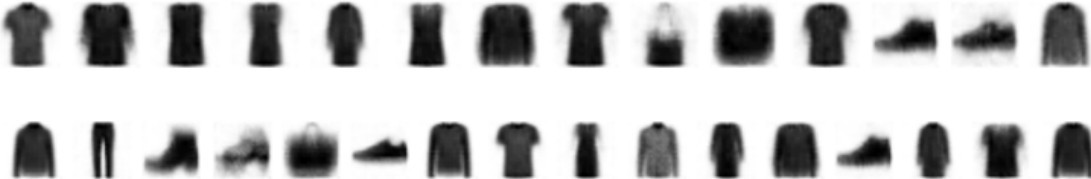

Option 2

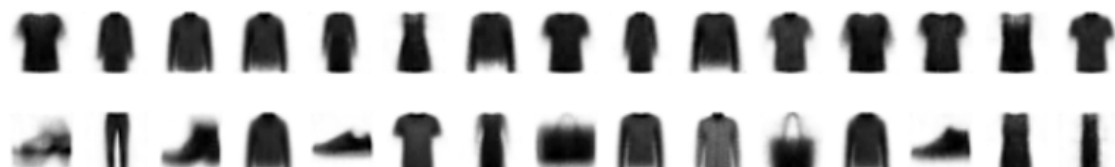

Option 3

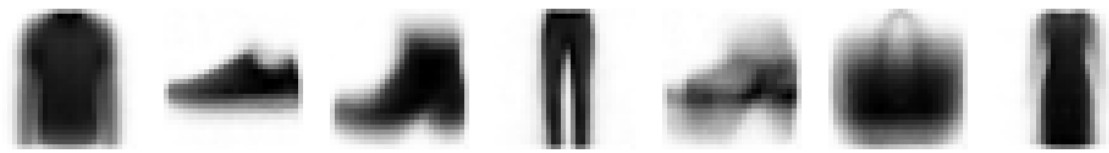

Option 4

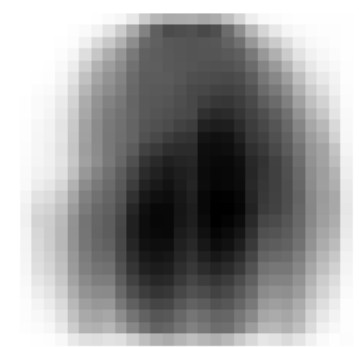

**Step 3:**

Please select which option you think best represents the groupings listed:

○ Option 1
○ Option 2
○ Option 3
○ Option 4

**Block 9**

# Debrief Page

Thank you for your participation, this study was designed to evaluate people's ability to select the right level of abstraction for an AI to perform well at a particular task.

if you have any questions please contact ekenny@mit.edu

Click next to finish

