

Figure 9: In experiments, we used a common feature-extractor ($F$), decoder ($D$), and predictor ($P$) network backbone while replacing different encoder heads ($E$).

# 7 Implementation details

Here, we include implementation details omitted from the main paper for brevity. Code for recreating the experiments in the paper is available at https://anonymous.4open.science/r/complexity_concepts-16A2. Upon acceptance, a deanonymized repository will be released.

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

 shared with participants of the user study, edited only to preserve anonymity during peer-review. Following standard user study procedure, we initial briefed users by telling them how long the study was and that they were free to leave at anytime. Demographic information was collected in person. After the study, users were debriefed and given the email of the study designer to contact if they had any questions.

**Block 1**

# Brief

You are free to leave this study at any time.

It will take less than 5mins.

Thank you for your participation!

**Block 2**

# Introduction

A factory has created a robot that is good at sorting clothes into these 10 categories

- T-shirt
- Trouser
- Pullover
- Dress
- Coat
- Sandal
- Shirt
- Sneaker
- Bag
- Ankleboot

See below for examples of these 10 categories

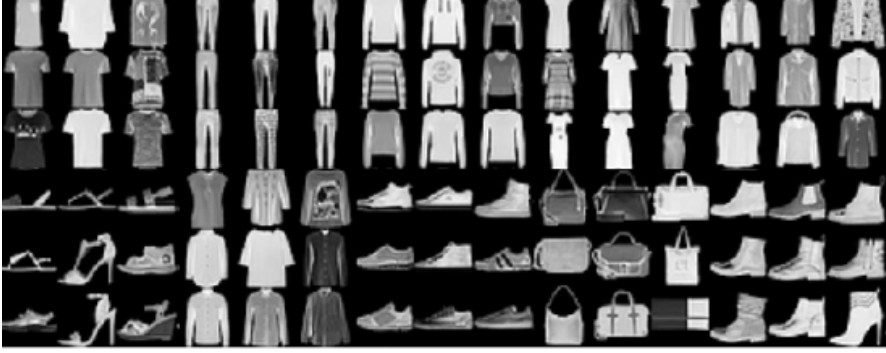

**Block 3**

# Introduction

However, in your home you don't care about these 10 categories.

You only sort your clothes into these 3 categories:

- Shirts
- Shoes
- Trousers/Bags

So, we want to teach the robot to be more general and only care about these three high level categories.

**Block 10**

# Introduction

In doing so, we are going to re-train **_two different robots_** over a period of time.

At a certain point during this re-training, they will be able to sort these clothes into these three new categories best.

**_Your task is to pick the point in which they are going to categorize these three high-level categories best._**

**Block 4**

# Introduction

You will have to answer one question about each robot.

**First**, you will be shown one robot which only communicates with its error score. *The lower the score, the better the robot is at classifying the 10 categories*.

**Secondly**, you will be shown another robot. This robot has learned to communicate what it has learned though visualizing the most important images it uses for categorizing the 3 high-level concepts. You have to decide which visualization corresponds to the robot being able to categorize the three new high-level categories best.

*Generally, the more images the second robot is using, the less general it will be, and the worse it will do at categorizing the 3 new high-level categories.*

**Block 5**

# Robot 1

Here is an example of the first robot communicating its error scores over training.

You will have to pick a score from 3 sampled options you think will perform best on your desired task of learning the 3 clothing concepts. *A lower score is typically seen as better (which corresponds to a lower point on the blue curve below).*

*Your task is to pick the point which you think will perform best at representing these three categories.*

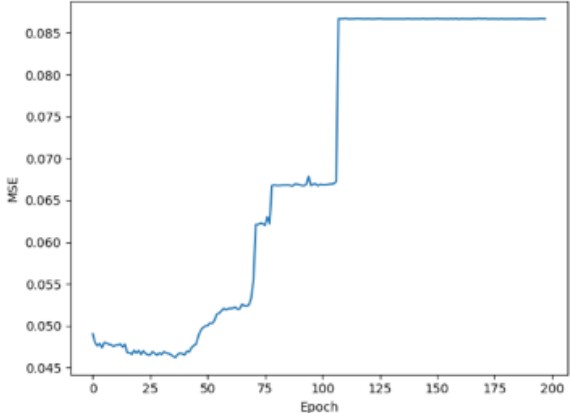

**Block 6**

# Robot 2

Then you will be shown visualizations like this, which is second robot's method of communicating its learned concepts.

If the visualization shows that (1) the robot is not using many images, and (2) they roughly represent the three high-level categories (shirts, shoes, trousers/bags), then the robot should perform well in the task of sorting clothes into these three categories.

***Your task is to pick the visualization which you think will perform best at sorting the clothes into these 3 high-level categories.***

For example, here is one robot's visualisation which is perhaps too general, as there is not even three categories being used.

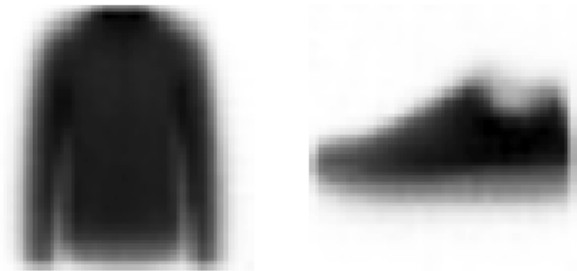

And here is one which could be too specific, because there is a lot of detail.

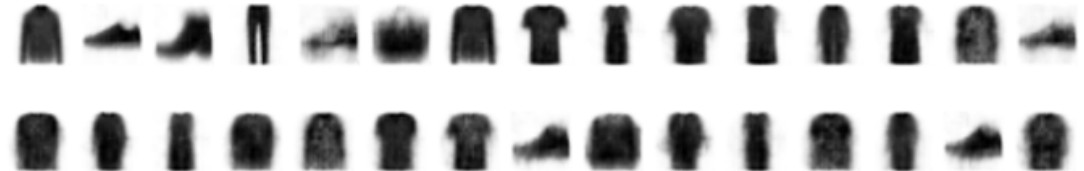

***Important: We want the robot to learn the three high-level concepts, but no more or no less!***

**Block 11**

# Click next to begin the study

**Block 7**

## Question 1:

Each circled region represents the robot trained with a different error score. Remember this error is on the 10-way clothing classification task.

 Choose the point that you think the robot is going to perform the best at classifying the high-level categories.

- Shirts
- Shoes
- Trousers/Bags

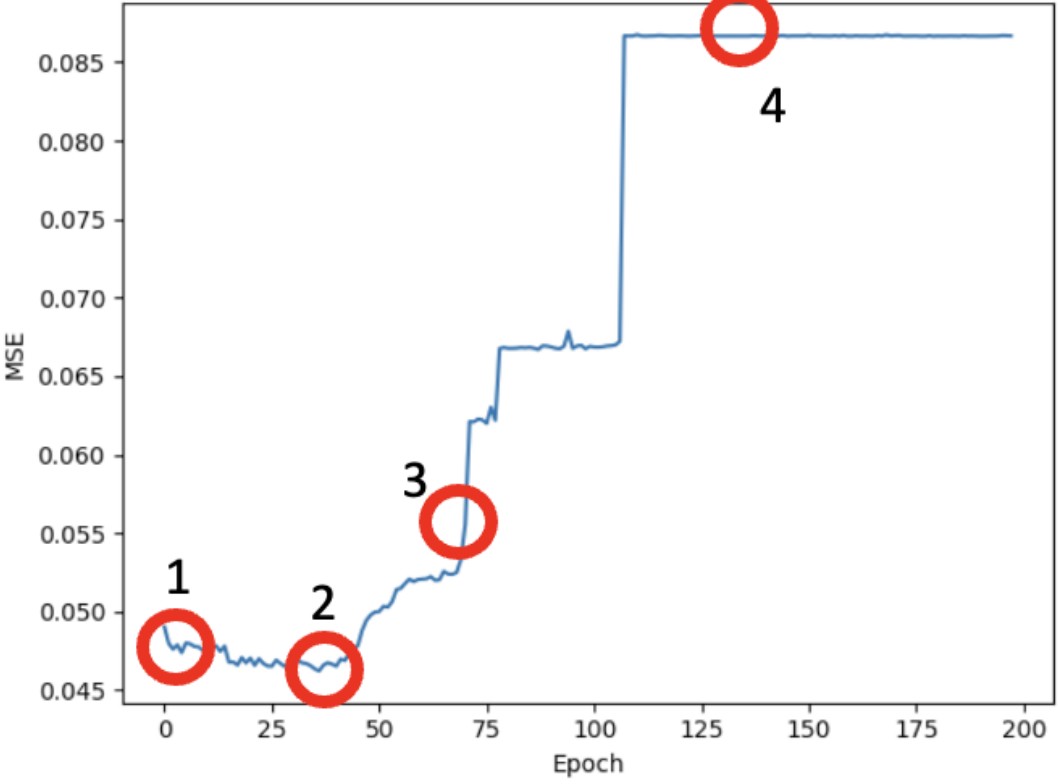

Choose the robot now:

○ 1
○ 2
○ 3
○ 4

**Block 8**

# Question 2:

Each visualization represents the robot trained with different high-level categories.

Choose the visualization that you think will help the robot perform best at sorting your clothes into these three high-level categories:

- Shirts
- Shoes
- Trousers/Bags

The first option is

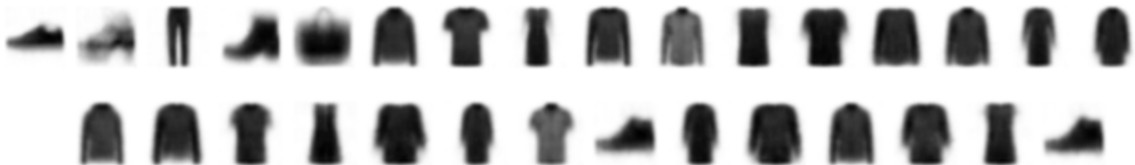

The second option is

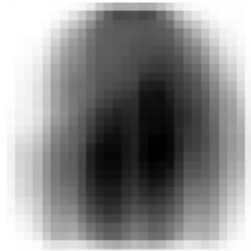

The Third Option is

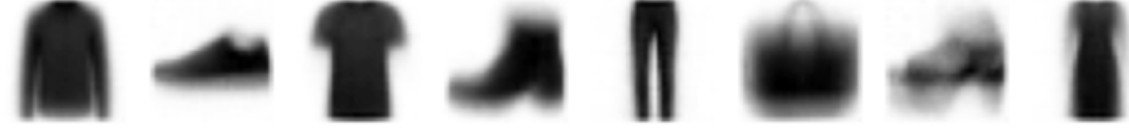

The Fourth option is

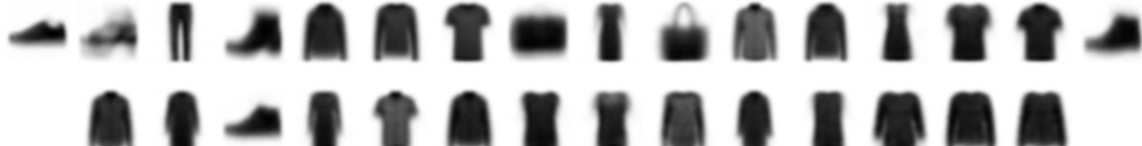

Click to write the question text

○ First option
○ Second option
○ Third option
○ Fourth option

**Block 9**

# Debrief Page

Thank you for your participation, this study was designed to evaluate people's ability to use an explainable artificial intelligence system to help fine-tuning towards down-stream tasks.

if you have any questions please contact 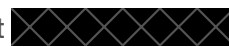

 Click next to finish

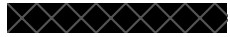