# OpenReview forum: "Human-Guided Complexity-Controlled Abstractions"
_NeurIPS.cc/2023/Conference — NeurIPS 2023 poster_

### Official Review · Reviewer_TMXo · 2023-07-02

**Soundness:** 3 good
**Presentation:** 3 good
**Contribution:** 3 good
**Rating:** 6
**Confidence:** 5

**Summary:**

This paper proposes a method for learning discrete representations whose complexity can be smoothly annealed. Experiments demonstrate that, at the appropriate level of complexity, these representations can be useful for downstream classification tasks involving abstract categories.

**Strengths:**

- The work proposes an interesting method for learning a discrete 'codebook' with a controllable level of complexity.
- Experiments demonstrate a clear non-monotonic relationship between complexity and usefulness in a downstream task involving abstract categories.
- The method outperforms baselines in this setting.

**Weaknesses:**

I have previously reviewed this work, and it seems that the paper has not been revised in a manner that sufficiently addresses the concerns raised by myself and other reviewers. Specifically, a major concern with this work as currently formulated is that the proposed human-in-the-loop approach does not seem very realistic. In the abstract, the authors note that 'humans learn discrete representations ... at a variety of abstraction levels ... and use the appropriate abstraction based on tasks'. This is a compelling motivation, but unlike humans, the proposed approach does not involve any method for autonomously selecting the appropriate level of abstraction for a given task. It is unclear how this human-in-the-loop approach will be scaled in a realistic manner, and the authors do not devote enough attention toward addressing this limitation.

It would help if the authors could add more discussion of this issue, and also describe more concretely the scenarios in which they envision this approach being useful. Given pre-trained models over a range of complexity levels, and some downstream task on which a user wants to fine-tune, would it not be much more straightforward to simply use a validation set from the downstream task to evaluate the full set of pre-trained models? This also suggests the possibility of automating the process of selecting the right abstraction level in a meta-learning setup.

**Questions:**

I would appreciate if the authors could devote more attention to the limitations identified in the previous section, and also provide more concrete detail about the intended use case for the proposed approach.

**Limitations:**

There are no discernible negative societal impacts related to this work.

---

> ### Author Rebuttal · Authors · 2023-08-03
>
> We thank the reviewer for their continued engagement with our work!
>
> >This is a compelling motivation, but unlike humans, the proposed approach does not involve any method for autonomously selecting the appropriate level of abstraction for a given task.
>
> The reviewer is correct that we do not provide a method for autonomously selecting the appropriate level of abstraction for a given task. Fundamentally, we propose a human-in-the-loop framework rather than an autonomous selection framework.
>
> Nevertheless, we appreciate the suggestion to regularize the finetuning models, using a validation set to tune regularization. As noted in our general response, we conducted such experiments and found that 1) there is a small benefit to regularization but 2) our main trends hold, such that tuning to the right complexity level supports the greatest finetuning accuracy, therefore resulting in our method still outperforming the (stronger) regularized baseline.
>
> We discuss an additional method for autonomously selecting the right encoder based on the number of prototypes in our overall response. This method, too, fails for both theoretical and empirical reasons.
>
> Our findings, with strengthened baselines and methods for choosing models based on the number of prototypes, establish the importance of our human-in-the-loop framework rather than autonomously selecting models. Ultimately, given the positive results from our human-in-the-loop framework, and our negative results from various autonomous selection baselines, we hope that the reviewer will reconsider their score.

---

> > ### Comment · Reviewer_TMXo · 2023-08-11
> > **Response to rebuttal**
> >
> > Thanks to the authors for these clarifications and additional experiments. The regularization experiments are interesting, but that is not what I had in mind when I mentioned fine-tuning with a validation set. What I meant was that, given some downstream task, and given models with a range of complexity, a natural solution would be to generate a validation set for evaluating those models based on accuracy for the downstream task. That seems like a much more straightforward solution than having a human look at the prototypes generated by the model, but perhaps I am missing something.
> >
> > I also still feel that the paper needs to more clearly acknowledge the limitations of the human-in-the-loop paradigm, and the need to develop a method for autonomously selecting the right encoder. The paper does not really discuss these issues, and the authors have not indicated that they will revise the paper to include such a discussion.

---

> > > ### Author Response · Authors · 2023-08-12
> > > **Followup experiments; reframing in overall response**
> > >
> > > # Paper Reframing
> > > In light of the discussion and reviewer’s recommendations, we have proposed a reframing of the paper. Please see the general comment above!
> > >
> > > # Experiment: validation set
> > >
> > > We thank the reviewer for the clarification on validation set experiments. We have now conducted such experiments and report our approach and the results below:
> > >
> > > Using the suite of pre-trained encoders from the original paper, for each $k$, we generated a train-validation split by holding out a validation set, $v$, from the trainset. Predictors were trained on the remaining training data and evaluated on the validation set. We conducted 5-fold cross-validation for different train-validation splits to record the average validation accuracy. We then selected the encoder with the highest average validation accuracy and finetuned it on both the train and validation data. We then recorded the test-set accuracy of these “optimal” models. We repeated this experiment over 5 random seeds (corresponding to the 5 pre-training runs).
> > >
> > > Results, averaged across the 5 random seeds, from our experiments are included in the tables below (for VQ-VIB$\_\mathcal{C}$). (Given the long duration of these cross-validation experiments, we are still waiting for the iNat results for $|v| = 5$ and will report them upon completion. So far, trends across all domains have been consistent.)
> > >
> > > ## FashionMNIST
> > > | $k$  	| 2 | 5 | 10 | 50 |
> > > | ----------- | ----------- | ----------- | ----------- | ----------- |
> > > | $\|v\|=1$  	| 0.73 | 0.89 |  0.93 | 0.97 |
> > > | $\|v\|=5$  	| - | - |  0.94 | 0.97 |
> > > | $\|v\|=10$  	| - | -| -| 0.97 |
> > >
> > >
> > > ## CIIFAR100
> > > | $k$  	| 2 | 5 | 10 | 50 |
> > > | ----------- | ----------- | ----------- | ----------- | ----------- |
> > > | $\|v\|=1$  	| 0.71 | 0.79 |  0.83 | 0.90 |
> > > | $\|v\|=5$  	| - | - | 0.82 | 0.91 |
> > > | $\|v\|=10$  	| - | -| -| 0.91 |
> > >
> > >
> > > ## iNat
> > > | $k$  	| 2 | 5 | 10 | 50 |
> > > | ----------- | ----------- | ----------- | ----------- | ----------- |
> > > | $\|v\|=1$  	| 0.65 | 0.79 | 0.89 |0.90 |
> > > | $\|v\|=5$  	| - | - |   | |
> > > | $\|v\|=10$  	| - | -| -| 0.89|
> > >
> > > With enough finetuning data, and a large enough validation set, this method indeed recovers near-optimal encoders. However, as in many of our experiments, in the small-data regime, this method remains suboptimal. For example, for FashionMNIST, $k=2$, the selected model on average achieves 73% accuracy; lower than the peak of roughly 80%. We suspect that this method fails in low data regimes because performance on a small validation set may not reflect performance on the test set, so using a validation set for selection only works with a large enough validation set. Indeed, with small validation sets, this method sometimes selected high-MSE models which happened to perform well on the validation sets but which performed sub-optimally on the test set.
> > >
> > > We thank the reviewer again for suggesting this technique.

---

> > > > ### Comment · Reviewer_TMXo · 2023-08-16
> > > > **Reply**
> > > >
> > > > Thanks to the reviewers for running these additional experiments. I think these results, combined with the proposed reframing, do strengthen the paper, so I am willing to increase my score to a 6. I still feel that the human-in-the-loop setting is not sufficiently motivated, as the model performs just as well autonomously with only ~10 fine-tuning examples and ~10 validation examples, which in general should be very easy to obtain.

---

### Official Review · Reviewer_Nw28 · 2023-07-07

**Soundness:** 4 excellent
**Presentation:** 3 good
**Contribution:** 2 fair
**Rating:** 6
**Confidence:** 3

**Summary:**

This paper proposes a framework for human-in-the-loop training of machine learning models where humans select among pretrained models with different complexity levels based on prototypes. The authors demonstrate that finetuning performance is significantly impacted by representation complexity in the experiments considered. Moreover, a user study demonstrates that humans are relatively successful at helping choose the correct complexity level for finetuning.

**Strengths:**

- The idea of using humans to help select abstractions for a given problem is an interesting idea for human in the loop ML.

- The proposed VQ-VIB approach seems like a reasonable implementation for this use case.

- There seems to be solid empirical verification of the very intuitive connection between representation complexity and fine-tuning performance.

**Weaknesses:**

- The authors only show transfer performance within single benchmarks rather than across tasks or in realistic pretraining and finetuning settings such as ImageNet or LLMs.

- The authors do not compare with other methods for human in the loop training in which the amount of human effort can be compared. For example, feature engineering is another way that humans can potentially impact downstream performance -- a technique widely used in practice.

- The paper also does not discuss the effort involved in providing this human guidance.

**Questions:**

1. Can you quantify the amount of effort needed by human annotators to select the best abstraction level?

2. Can you quantify the amount of sub-optimality in the human study when the correct abstraction was not selected?

3. How does the effort and efficacy involved in prototype based abstraction selection compare with that of feature engineering?

4. There is something off about the bird abstraction example (line 20). What precisely is the task the abstraction is being used for in this case? Is the expert birdwatcher teaching the child or other experts? What does complexity and simplicity mean in this case?

**Limitations:**

The authors do not really address the limitations of the work. I believe the main limitations are the lack of alternative methods for human-in-the-loop training considered i.e. feature engineering and the use of somewhat synthetic benchmarks for evaluation that are disconnected from real-world use cases. It is natural to wonder if humans can be as helpful at prototype based abstraction selection for transfer tasks that are more distant from each other and not drawn from the same benchmark.

---

> ### Author Rebuttal · Authors · 2023-08-04
>
> >The authors only show transfer performance within single benchmarks rather than across tasks or in realistic pretraining and finetuning settings such as ImageNet or LLMs.
>
> While we do not consider LLMs (we did not wish to expand to language domains), we do use feature extractors pre-trained on ImageNet (as noted on line 457).
>
> More generally, we do not seek to make any claims or consider transfer to different domains. Cross-domain transfer is an interesting idea, but here we seek to answer the first question of finetuning efficiency within a single domain, with the goal of understanding how to best develop visual representations that are adaptable and flexible for different end tasks.
>
> >The authors do not compare with other methods for human in the loop training in which the amount of human effort can be compared. For example, feature engineering is another way that humans can potentially impact downstream performance -- a technique widely used in practice.  Can you quantify the amount of effort needed by human annotators to select the best abstraction level? How does the effort and efficacy involved in prototype based abstraction selection compare with that of feature engineering?
>
> While comparison to other human-in-the-loop pretraining methods is an interesting (and novel!) space to explore, we believe our method exists orthogonal to traditional “manual feature engineering”. First, feature engineering typically requires the designer to have full knowledge of the desired downstream task, effectively requiring *a priori* knowledge of the test task during pretraining (e.g. we need to first know that we are trying to classify birds to identify that beaks are important) [1,2,3]. A specific model is then trained to fit those features. In contrast, our method assumes that we do not possess this information (nor access to the end user during pre-training), but rather “fits” the right representation adaptively at test time. Second, traditional human feature engineering is known to be labor and expertise-expensive [1,2], often requiring either designer knowledge of the feature extraction space or specific visualization tools designed to extract this information [3]. For example, in order for us to run this comparison in line with [3], we would have needed to design a web-interface tool for FashionMNIST, CIFAR100, and iNat, and then have brought in human participants to design features for each downstream task we desired, which would be extremely time consuming (for both the designer and participants).
>
> We apologize that we did not report results on the time/ease of our human experiment and will correct this omission. Our method averaged ~1m30s per human participant, INCLUDING TEACHING TIME, for each question. This required no designer knowledge of the test task, nor time spent deploying visualization tools or interfaces for extracting end user knowledge of desired features – we simply needed to output the representations that were already generated during pretraining. This illustrates the relative ease and flexibility of deploying our method across different tasks with different end users.
>
>
> >Can you quantify the amount of sub-optimality in the human study when the correct abstraction was not selected?
>
> Our computational results, in conjunction with the human study, indicate the suboptimality of choosing the incorrect abstraction. For example, when users selected the lowest MSE model, that corresponded to accuracy of 55\%, as opposed to 70\% accuracy when selecting the best model (via our method), for $k=1$. (Percentages taken from Figure 4c in the main paper). One can generally quantify suboptimality by looking at the options selected by participants and then looking up the accuracy of such options in our computational experiments.
>
> >There is something off about the bird abstraction example (line 20). What precisely is the task the abstraction is being used for in this case? Is the expert birdwatcher teaching the child or other experts? What does complexity and simplicity mean in this case?
>
> We describe two tasks: an expert classifying birds (line 23) or teaching a child to classify birds into crude categories (line 24). In this context, complexity corresponds to the level of detail about the bird used for the task (e.g., specific plumage patterns for exact species classification vs. general color for the simpler task). Thus, we expect the expert to use a complex representation for classifying species and a less complex representation for discussing high-level colorings with children.
>
> [1] "An empirical analysis of feature engineering for predictive modeling." Heaton.
>
> [2] "Runtime Support for Human-in-the-Loop Feature Engineering System." Anderson et al.
>
> [3] "Get a human-in-the-loop: Feature engineering via interactive visualizations." Gkorou et al.

---

> > ### Comment · Reviewer_Nw28 · 2023-08-15
> > **Response to Rebuttal**
> >
> > Thank you for your detailed responses to my questions. This definitely helped clarify with respect to some of my concerns. In particular, it was quite interesting to know about the average time taken by the human participants. This was lower than what I would have expected. As a result, I have decided to raise my score after the rebuttal.

---

### Official Review · Reviewer_99tN · 2023-07-07

**Soundness:** 3 good
**Presentation:** 3 good
**Contribution:** 2 fair
**Rating:** 5
**Confidence:** 3

**Summary:**

The authors observe that the downstream tasks for pretrained models can rely on representations of varying level of complexity: as a running example, a birdwatcher relies on significantly more complex representations of images to classify bird species, relative to a child who may want to identify the color of a bird. They thus propose pretraining a variety of models with representations of varying complexity, having a human choose the appropriate pretrained model for the task, and then finetuning that model for the downstream task. Relative to finetuning the model with the most complex representations, this should lead to increased data efficiency.

The authors suggest a modification of VQ-VIB (vector-quantized variational information bottleneck) for the pretraining in this setting, called VQ-VIB_C (C standing for categorical) – my understanding is that the main difference is that in the authors’ method, the representation is divided into n chunks and each of the n chunks is snapped to a quantized vector (the loss remains the same). I have not spent much time delving into the details as the authors say this is not their main contribution: in particular I may be wrong about the differences from VQ-VIB.

The authors implement this idea for a variety of image-based datasets: FashionMNIST, CIFAR-100, and iNaturalist 2019 (iNat). They pretrain an encoder using a reconstruction loss as well as a classification loss for the labels defined by the dataset, and increase the regularization on the representations to generate a variety of checkpoints that produce representations of different complexity. They then evaluate these encoders in a downstream classification task that is coarser than the original classification task (e.g. living vs. non-living for CIFAR-100).

Their experiments show that (1) VQ-VIB_C outperforms two baselines (VQ-VIB, $\beta$-VAE), and (2) when the classification task is simple (e.g. 2-3 classes) and there are very few data points (e.g. 1-5 examples per class), it is better to use a pretrained encoder with lower representational complexity (though in all other cases it is typically fine to use the encoder with maximal representational complexity).

They also perform a human experiment in which they show that, given visualizations of the learned quantized vectors, humans can correctly identify which encoder will perform best during finetuning.

**Strengths:**

1. To my knowledge, the idea of using simpler representations for downstream tasks in order to improve data efficiency is novel, and I think it is an interesting hypothesis to explore.
2. Although the authors don’t consider it their main contribution, their experiments suggest that the VQ-VIB_C performs notably better than other alternatives on their task. (However, I did not investigate in detail, e.g. it is possible that the baselines were not tuned well while the VQ-VIB_C was.)
3. More generally, the experiments are quite detailed and look at the effects of various hyperparameters in the method.
4. The paper has an experiment with real humans to justify its claim that humans could choose an appropriate model to use for a downstream task. While I think the experiment is pretty different from realistic conditions, this is still better than the vast majority of papers on ideas like these, which often don’t bother testing their claims about real humans at all.

**Weaknesses:**

I have a few concerns:
1. Significance: It seems to me that the author’s experiments suggest that the main idea only has a benefit in very limited settings, which are unlikely to arise in practice.
2. Baselines: For k > 1, the appropriate baseline would be to use the most complex representation, with strong regularization determined using a validation set at finetuning time.
3. External validity: The experiments involve downstream tasks that are pure coarsenings of the pretraining task, whereas in typical settings the downstream tasks will likely not be these “pure coarsenings”, which would reduce performance.
4. Relevance of the human experiment: The author’s experiment seems to “give away” the answer (though there is a case to be made that this reflects a strength of the method, rather than a weakness of the experiment).

Overall I feel conflicted about this paper. On the one hand, it has a nice idea with careful technical work done to flesh it out, and a large set of experiments to understand how the idea works in practice. On the other hand, I think the experiments suggest that the idea is not very practically useful (whereas the paper suggests they validate the idea), despite the experiments having some aspects that bias them towards showing the idea to be good. Overall I’m recommending a borderline reject, but I can see the case for acceptance as well.

**Significance**

Looking at the experiment results (including the ones in the appendix), it appears that if n > 3, or k > 5, or you use any method other than VQ-VIB_C, it is usually best to use the most complex representation (i.e. the one that achieves highest reconstruction loss). Thus the benefits of the idea only occur when n <= 3 and k <= 5, which implies a tiny dataset of 15 examples or fewer. As we might expect from such a small dataset, the resulting finetuned classifiers do not perform particularly well, getting around 70-90% accuracy.

Thus it seems like the idea in this paper is only helpful when (1) there is a very simple classification task, (2) there is very limited finetuning data available, (3) you use VQ-VIB_C rather than a different method, and (4) the user is happy with performance numbers of 70-90% accuracy. This seems extremely restrictive, and I find it hard to think of a realistic setting that satisfies all of these constraints (especially #4). Overall, I would characterize these experiments as providing a negative result.

(I would actually be more inclined to accept a version of this paper that was upfront about this, and framed the paper as a negative result that has taught us that lower representation complexity only buys you a little bit of data efficiency, that is overwhelmed very quickly by a large enough dataset.)

**Baselines**

In the paper’s experiments, the finetuning is done the same way for representations with varying levels of complexity (I believe, at least I didn’t see anything to contrary in Appendix 7). However, a natural baseline would be to use the regular (maximum-complexity) representations, but use stronger regularization to learn a simpler classifier. One might reasonably argue that it is unclear how to choose this hyperparameter – but at least when k > 1, it should be possible to take 1 example (or more) per class to form a validation set that is used to tune the hyperparameter. I think it is plausible that this significantly improves performance for high-complexity representations. If it does work it would be a significant improvement, as there would no longer be any need for human input.

**External validity of experiments**

All of the experiments in the paper have the downstream task labels $Y_t$ being a strict coarsening of the pretraining task labels $Y_p$ (or in other words, $Y_t$ is a deterministic function of $Y_p$). However, this may not be the case in realistic settings – for example, in the running example, there may be birds that are of the same species but have different colors, and so a representation that just identifies the species would be insufficient for predicting the color. Should we expect the method to work even in these cases? I’m not sure; the fact that the pretraining includes a reconstruction loss suggests that it would still work (though likely not as well as in the paper’s experiments). Ideally however the authors would conduct such an experiment to test it empirically.

**Relevance of the human experiment**

Looking at the pdf for the user study (Appendix 11), the instructions seem to “give away” the answer. In particular, the instructions contain:

> Generally, the more images the second robot is using, the less general it will be, and the worse it will do at categorizing the 3 new high-level categories.

and

> If the visualization shows that (1) the robot is not using many images, and (2) they roughly represent the three high-level categories [...] then the robot should perform well

and

> For example, here is one robot’s visualization which is perhaps too general, as there is not even three categories being used.

From which it is easy to infer “choose the option with fewest categories, subject to having at least three categories” – which then leads to the desired answer.

Arguably this represents a strength of the method – the approach for selecting the appropriate model is so simple that it can easily be communicated even to non-experts. However, in this case, it is so simple that there isn’t really any need for human input – we could equally well find the appropriate model using a simple heuristic of choosing the option that has a few more “most important images” than the number of classes in the classification task. (Perhaps this is another baseline which should be compared against.)

**Minor issues**

Line 91:
> Lastly, we assume that the pre-training labels are a sufficient statistic for the task-specific labels: $I(X; Y_p) = I(X; Y_t)$. Intuitively, this states that the pre-training objective must include relevant information for the finetuning task.

I don’t think that’s the right criterion. For example, for reconstruction pretraining with red-yellow classification as the downstream task, we have $Y_p = X$ and $Y_t$ is whether the bird is red or yellow. Then $I(X; Y_p) = I(X; X) = H(X) \geq 1$ (since entropy of natural bird images is large), but $I(X; Y_t) \leq H(Y_t) \leq 1$ (since $Y_t$ is a binary variable).

I think what you mean to say is that $I(X; Y_t \mid Y_p) = 0$, that is, conditioned on knowing $Y_p$, there is no more information about $Y_t$ that can be learned from $X$.

**Questions:**

1. Could you give examples of realistic settings in which using low-complexity representations would be useful – bearing in mind the four requirements I mention in the “Significance” section in Weaknesses above? (Or alternatively, if I’m wrong about the four requirements, how / why am I wrong?)
2. Is there reason to expect poor performance from finetuning the most complex representation with stronger regularization? (Possibly with a validation set, as described above)
3. How well should we expect the method to work when $Y_t$ is not a deterministic function of $Y_p$?

Minor suggestion: Please increase the size of the text in Figure 4(c).

**Limitations:**

The limitations brought up in the weaknesses section are not mentioned or addressed. In particular, I think the issues raised in “Significance” and “External validity of experiments” should obviously be mentioned (or otherwise addressed); the others are more debatable.

---

> ### Author Rebuttal · Authors · 2023-08-04
>
> We thank the reviewer for their helpful review. They understood the paper quite well and provided useful suggestions for strengthening baselines (which we implemented).
>
> # Significance
>
> >[T]he benefits of the idea only occur when n <= 3 and k <= 5… classifiers do not perform particularly well, getting around 70-90% accuracy. Thus it seems like the idea in this paper is only helpful when [4 constraints are met]…
>
> The reviewer is correct that for every architecture other than VQ-VIB$_\mathcal{C}$ (our method), we see no benefit to selecting a less complex representation. However, it is not true that “the benefits of the idea only occur when n <= 3 and k <= 5.” For example, for FashionMNIST, performance peaks at the “right” complexity level even for $k = 50$ (Figure 10 in Appendix 8). Notably, this peak supports roughly 96% accuracy.
>
> We acknowledge that the advantages of our method are limited to the small-data regime. However, rather than frame the whole paper as a negative result, we wish to emphasize that while some existing methods ($\beta$-VAE, VQ-VIB$\_\mathcal{N}$) do not confer benefits from simpler representations, VQ-VIB$\_\mathcal{C}$ does! In low-data scenarios, our method demonstrates its value over existing techniques.
>
> # Limitations
>
> >I think the issues raised in “Significance” and “External validity of experiments” should obviously be mentioned.
>
> We note limitations of our approach as the amount of finetuning data increases (line 167) and repeat this conclusion throughout Appendix 8 (line 631). In future revisions, we will expand such discussion.
>
> # Questions
>
> >Could you give examples of realistic settings in which using low-complexity representations would be useful – bearing in mind the four requirements I mention…? (Or alternatively, if I’m wrong about the four requirements, how / why am I wrong?)
>
> We believe that the real requirements are less strict than noted by the reviewer: we found benefits to our method when finetuning for FashionMNIST using 50 examples, and achieved 96% accuracy. Thus, requirements 2 (limited finetuning data, k <= 5) and 4 (performance under 90%) are not supported by our results. We also do not see how using VQ-VIB$_\mathcal{C}$ is a limitation: it outperforms baselines and therefore is the method that we advocate for in this work.
>
> We believe that our approach could be relevant for low-volume visuomotor robotics settings. Considerable research considers how to rapidly train robots on sorting tasks from vision using few demonstrations, where minimizing the number of new demonstrations required at test time from the human user is desirable [1,2,3]. Using our approach, we could train a robot to identify clothing for sorting (as in our FashionMNIST example) using only 10 examples per class and achieve roughly 93% accuracy! If a factory wanted to retrain the robot on a different sorting task (e.g., a binary defect-identification task), it could quickly supply the small number of finetuning examples needed. This type of flexible test-time adaptation motivates our framework.
>
> >Is there reason to expect poor performance from finetuning the most complex representation with stronger regularization?
>
> We appreciate the suggestion to train with stronger regularization and did so! Results, as indicated in our overall response, are included in the attached pdf.
>
> We found that regularization improved model performance overall, but there was nevertheless still the “peaking” effect of optimal performance when tuned to the right complexity level. Intuitively, this arises because regularization can help address aspects of domain complexity, but fundamentally it is still easier to train with less complex domains. Thus, our method continues to outperform the strong baseline of training a regularized classifier on the most complex representation.
>
> > (Paraphrased:) What if the finetuning task labels are not a deterministic coarsening of the pre-training labels?
>
> This is an interesting question which we explicitly flagged as outside the scope of our current paper. (“Lastly, we assume that the pre-training labels are a sufficient statistic for the task-specific labels…”) Nevertheless, we can speculate what behavior we may see.
>
> We believe that, if the finetuning labels are less aligned with the pre-training labels, we would see peaked finetuning performance at higher complexity (lower MSE) than in our current experiments (or no peak at all). Generally, compressing model representations helped with finetuning, but only up to the point where information necessary for the finetuning task started being compressed out. With worse pre-training labels, we might expect models to start compressing information that is important for finetuning earlier. Thus, peak performance should occur at higher complexity values.
>
> > (Paraphrased:) Could one automatically select the right complexity level based on the number of prototypes?
>
> We discuss this question in our overall rebuttal. While an interesting hypothesis, there are theoretical reasons that this approach should fail, which we back up in experiments.
>
> # Minor issues:
>
> Thank you for pointing out the error. We wrote the correct text, “we assume that the pre-training labels are a sufficient statistic for the task-specific labels” but miswrote the equation. We will correct the document to write $I(X, Y_t) = I(Y_p, Y_T)$.
>
> [1] "Robot learning from randomized simulations: A review.” Muratore et al.
>
> [2] "A Methodology to Design and Evaluate HRI Teaming Tasks in Robotic Competitions" Marrella, Andrea, et al.
>
> [3] “Preferred interaction styles for human-robot collaboration vary over tasks with different action types” Schulz et al.

---

> > ### Comment · Reviewer_99tN · 2023-08-12
> > **Thanks for the response**
> >
> > I have read through the authors' rebuttal and the other reviews, and am maintaining my score of 4.
> >
> > ## Baselines
> >
> > Thanks for running the updated baseline! I'm more convinced now that the method shows benefits in the very-low-data regime ($n \leq 3, k \leq 5$).
> >
> > ## Significance
> >
> > I am still not convinced that the method shows benefits for $k > 5$. Looking at $k = 50$ in Figure 10 in Appendix 8, the curves of best fit show a peak at non-maximal complexity, but if you ignore the curves of best fit and just look at the actual data, it is far from clear that there is an actual difference between maximal complexity vs. the point at which the curve of best fit peaks. If there is a difference, it is very small in magnitude. If we instead look at the version with updated baselines (lightest green points in Figure 24a in the author rebuttal pdf), there is even less of a visible difference. To me, these graphs suggest that there is a plateau of maximal performance that includes the maximal complexity representation.
> >
> > However, even if we grant that Figure 10 and Figure 24a show an improvement in the $k = 50$ case for 3-way FashionMNIST, that is a cherrypicked example. If we instead look at Figures 24g / 13 (iNat 3-way), Figure 12 (CIFAR100 20-way), Figure 14 (iNat 34-way), Figure 15 (iNat 1010-way), we see that the best choice at $k = 50$ is the maximal complexity representation. Only Figure 11 / 24d (CIFAR100 2-way) show a peak at non-maximal complexity, and those are even more arguable than Figure 10 and 24a.
> >
> > (Incidentally, I don't trust the curves of best fit. My guess is that the authors are fitting cubic polynomials to the data? When one fits a cubic to a set of data that increases and then saturates / asymptotes creating a plateau, the cubic will tend to put its maximum value in the middle of that plateau -- thus creating an illusion that the maximum value is in the middle of the plateau, rather than at the end of it.)
> >
> > ## Constraints and realistic settings
> >
> > > Using our approach, we could train a robot to identify clothing for sorting (as in our FashionMNIST example) using only 10 examples per class and achieve roughly 93% accuracy.
> >
> > Yes, I would agree that this would be a plausible application. But currently, my sense is that at $k = 10$: (a) the method provides a very small boost (if any) relative to using the maximum complexity representation, and (b) this only happens in some of the settings tested (FashionMNIST), while in others (iNat) there is no boost. So overall this does not seem like a significant improvement, especially given the additional cost of human intervention.
> >
> > To put it a different way, the best case seems to be: "By default, we can train a robot to identify clothing for sorting using only 10 examples per class and achieve roughly 92% accuracy. With our method, we can get costly human input to select an appropriate representation, and then we can train it to 93% accuracy". And it is not clear to me whether even this is achieved, given the experiments in settings like iNat.
> >
> > > Reviewers 99tN and TMXo suggested the hypothesis that the optimal encoder is the one that uses approximately the same number of prototypes as finetuning labels. Such an approach does not succeed.
> >
> > I don't mean to imply that this holds in general. I am saying that your human experiment involved instructions that strongly suggested that the humans follow the rule "choose the encoder that has the minimum number of prototypes, but at least as many as there are finetuning classes", and that in the specific case studied in your human experiment this was the right answer. So I cannot tell from this experiment whether humans would choose encoders appropriately in more complicated settings, like the handbags with / without straps example you give.

---

> > > ### Author Response · Authors · 2023-08-12
> > > **Updated framing!**
> > >
> > > # Paper reframing
> > > Please see the general comment above for a proposed paper reframing!
> > >
> > > # Updated human experiment
> > >
> > > >  I cannot tell from this experiment whether humans would choose encoders appropriately in more complicated settings, like the handbags with / without straps example you give.
> > >
> > > Thank you for clarifying this concern! As written in the general response, we also conducted a follow-up user study for the FashionMNIST experiment where models were finetuned on an arbitrary 3-way classification task (such as the theoretical example posed). Groupings were intentionally non-meaningful (Group 1: Tshirt, Trouser, Sandal; Group 2: Pullover, Dress, Sneaker; Group 3: Coat, Shirt, Bag, Boot).
> > >
> > > In this domain, peak performance was achieved for models using far more than just 3 prototypes (and therefore the “choose the encoder that has the minimum number of prototypes, but at least as many as there are finetuning classes” heuristic fails). Using this novel 3-way grouping for FashionMNIST, our follow-up user study confirmed that participants were **still able to correctly identify the optimal abstraction level for this task**, indicating that humans are still able to correctly select the optimal encoder in more complicated settings when the heuristic does not hold. We are happy to present both human-validated tasks in the revision.

---

> > > > ### Comment · Reviewer_99tN · 2023-08-14
> > > > **Raising my score to 5**
> > > >
> > > > Thanks! I hadn't realized the experiment mentioned in the general response was a *user study*, I thought it was done in simulation. That does address my concerns.
> > > >
> > > > I also like the reframing of the paper.
> > > >
> > > > Overall with these changes I'm happy to raise my score from 4 to 5.
> > > >
> > > > (Note for other reviewers / the AC: I'm not as familiar with the appropriate baselines to compare against when VQ-VIB_C is presented as one of the contributions. Previously I was evaluating the paper from a lens where the main contribution was the human-in-the-loop approach to selecting appropriate representations, and VQ-VIB_C was one particular method to test the idea on. Still, there are comparisons to other algorithms in the paper, that seem like reasonable choices to me.)

---

### Official Review · Reviewer_6aXd · 2023-07-10

**Soundness:** 2 fair
**Presentation:** 1 poor
**Contribution:** 2 fair
**Rating:** 5
**Confidence:** 4

**Summary:**

This paper explores an interesting premise -- how does the level of abstraction captured by a discrete (visual) representation dictate downstream task performance, where downstream tasks can be at arbitrary levels of abstraction. Specifically, the running example from the work that I really like is that of bird-watching; when communicating amongst experts, the extremely specific species names (e.g., "white osprey") is very useful, and captures the right level of information. However, when communicating with small children who are looking at things in the wild, coarser descriptions (e.g., "the white bird" or "the red bird") are much more useful.

To study this premise, this paper makes two contributions: first, it introduces a new method for learning discrete representations at various levels of abstraction, the Vector-Quantized Variational Informational Bottleneck - Categorical (VQ-VIB_c). Second, and more importantly, it presents a thorough evaluation, including a human-in-the-loop user study (N = 17), showing conclusively for downstream tasks with "known" abstraction levels, the "best" representation to use is the one that roughly matches that level of abstraction.

**Strengths:**

This paper starts with a strong motivating question, introduces a new method to explore the hypotheses formed from this question, and critically performs a comprehensive evaluation and user study that confirms the proposed hypothesis. I think the user study linking "prototypes" from different models (capturing different levels of abstraction) to the downstream finetuning task's level of abstraction is the most convincing, and most meaningful result in this work. I really like this type of study, and it really helps support the paper's key contributions.


**Weaknesses:**

Unfortunately, I have several concerns with this paper, stemming from the very confusing presentation of the paper (it took me a very long time to understand the experimental protocol, and I found that several details about the pretraining/finetuning procedure were missing or placed in scattered places in the Appendix), to the actual useability of such an approach for more practical applications.

First, on the presentation/soundness side -- I ask this question more explicitly below, but what is the exact link between the process of "degrading" a high-accuracy model (pretrained to high accuracy on the "full" classification task), and emerging levels of abstraction? In other words, how exactly does the process of changing the loss hyperparameters lead to models that capture different levels of granularity? From my read of the paper/appendix, this link is never made explicit, and the process is further left underspecified -- if hyperparameters are changing in a staged way (or you're running for a limited number of additional gradient steps given the "high accuracy" model), couldn't a possible confound be the initial batches seen during this "degradation process"? Another confound could just be the per-example/per-class difficulty, which is known to be uneven across classes in the datasets (e.g., iNaturalist has severe label imbalance out of the box, papers from the Distributionally Robust Optimization literature report huge discrepancies in average vs. worst class accuracy for other datasets)? Is this controlled for in any way?

Beyond these questions, I find it unclear how such an approach would be useful for practical applications? Is the idea that given some supervised dataset, you should train all models to "best accuracy" then selectively degrade the resulting representations depending on your (family of) downstream tasks? Isn't this expensive/redundant?

---
EDIT (Post-Rebuttal): I think many of my soundness concerns have been addressed during rebuttal, and in discussion with the other reviewers. Raising my score to a 5.

**Questions:**

I am very confused at the two-phase approach for varying representational capacity during pretraining. The paper seems to indicate that at first, models are pretrained until they obtain perfect accuracy at the (full) classification problem (e.g., the 10-class classification in FashionMNIST, the 100-class classification in CIFAR10). Then, after that happens, by tweaking the loss hyperparameters (e.g., downweighting "utility" and upweighting "representation conciseness" / "commitment loss"), you compress the representations until they are not meaningful, and look at various saved models that are produced in the course of that process for downstream finetuning (hence the plots that track accuracy on downstream vs. MSE).

Why does this approach make sense at all for evaluating the "conciseness/abstraction" of a representation? How are you supporting the claim that the "trajectory of models" between high accuracy and "indistinguishable" representations is capturing different levels of abstraction in the representation space? This feels like a major major connection that needs to be made explicit in the paper.

**Limitations:**

The main body of the paper does not explicitly discuss limitations; the appendix discusses some of the limitation of the VQ-VIB_c approach in passing, but a proper treatment of the pros/cons of this approach is not present.

---

> ### Author Rebuttal · Authors · 2023-08-03
>
> We thank the reviewer for their review!
>
> # Clarifications
>
> The reviewer raises several questions about how we generate “degraded representations.” Here, we seek to clarify our approach with a brief summary and address specific questions later.
>
> First, we train a VQ-VIB$_\mathcal{C}$ model to high accuracy and low reconstruction loss, by setting high values for $\lambda_I, \lambda_U$ and a low value for $\lambda_H$. After convergence, we increase $\lambda_H$ by a small amount every epoch. By increasing $\lambda_H$, we imposing a higher penalty on the entropy of representations. Thus, as $\lambda_H$ increases, the model uses fewer representations, each of which represents more abstract concepts. The idea of varying a hyperparameter to induce different representation complexity levels is widespread in Information Bottleneck literature for continuous [1, 2] and discrete [3, 4, 5] representations.
>
> # Questions
>
> >What is the exact link between the process of "degrading" a high-accuracy model… and emerging levels of abstraction? … [H]ow exactly does... changing the loss hyperparameters lead to models that capture different levels of granularity?
>
> We increase $\lambda_H$ after convergence to penalize the entropy of representations, resulting in the model using fewer representations, which necessarily represent more abstract concepts. This in turn leads to decreased model accuracy. These links between hyperparameter tuning, representation complexity, and model accuracy have also been well established in prior literature, such as [3], who explicitly show that decreasing complexity leads to more abstract representations. Similar ideas are explored both theoretically [6] and empirically [1, 2, 4] in other works as well.
>
> We note in our paper that we increase $\lambda_H$ during pre-training (line 179) and show in experiments how representations change over the course of varying $\lambda_H$ (e.g., Figures 4, 7, 16, and 18).
>
> >If hyperparameters are changing in a staged way… couldn't a possible confound be the initial batches seen during this "degradation process"? Another confound could just be the per-example/per-class difficulty…?
>
> The reviewer makes an interesting point, but it appears unlikely that the suggested confounds had an effect in our experiments. We trained and evaluated all methods on the same classes and data, so class imbalance would be unlikely to favor one method over another. Furthermore, we trained across many random seeds and for many epochs, using traditional methods of loading randomly shuffled batches. Therefore, we believe the change in performance across models may be attributed solely to our manipulated variables, e.g. changes in $\lambda_H$ (for a given model type) or changes in architecture (e.g., comparing VQ-VIB$_\mathcal{C}$ to $\beta$-VAE).
>
> >I am very confused at the two-phase approach for varying representational capacity during pretraining... [M]odels are pretrained until they obtain perfect accuracy… after that happens, by tweaking the loss hyperparameters (e.g., downweighting "utility" and upweighting "representation conciseness" / "commitment loss"), you compress the representations until they are not meaningful...
>
> > Why does this approach make sense at all for evaluating the "conciseness/abstraction" of a representation? How are you supporting the claim that the "trajectory of models" between high accuracy and "indistinguishable" representations is capturing different levels of abstraction in the representation space?”
>
> The reviewer mostly summarized our pre-training process correctly, although we note that we only vary a single hyperparameter during training to penalize the complexity of representations (and not, as stated by the reviewer, by varying the commitment or utility weights). (See line 179.)
>
> We have already discussed above how we use $\lambda_H$ to penalize the entropy of model representations, which is motivated by substantial theoretical and empirical existing literature.
>
> We show in experiments that VQ-VIB$\_\mathcal{C}$ models indeed learn different levels of abstraction at different complexity levels during our pre-training process. We show examples of a model from a single training run at high and low complexity in Figure 4. We show a similar example in the iNat domain in Figure 7. Moreover, in Figures 16 and 18, we show how representations evolved during training (as we penalize complexity of representations) for VQ-VIB$_\mathcal{C}$ models. In each figure, we show how lower-complexity models (which differ from high-complexity models only by training with a greater $\lambda_H$) use fewer and more abstract representations. Lastly, in every plot with MSE as an x axis, we implicitly show that in pre-training, we forced models to go from low-MSE (high complexity) to high-MSE (high complexity) representations.
>
> > The main body of the paper does not explicitly discuss limitations… a proper treatment of the pros/cons of this approach is not present.
>
> We thank the reviewer for raising this point. We would like to highlight that we do include a discussion of limitations of our approach in the main paper (line 265), but will work to ensure they are emphasized more clearly in the future draft.
>
> We hope these clarifications help increase the reviewer’s understanding of our work, and we ask that the reviewer adjust their score if their confusion is sufficiently
> addressed, or suggest specific changes they would like to see to address their concerns.
>
> [1] Deep Variational Information Bottleneck. Alemi et al.
>
> [2] beta-VAE: Learning Basic Visual Concepts with a Constrained Variational Framework. Higgins et al.
>
> [3] Trading off Utility, Informativeness, and Complexity in Emergent Communication. Tucker et al.
>
> [4] Proto-VAE: A Trustworthy Self-Explainable Prototypical Variational Model. Gautam et al.
>
> [5] Representation Learning in Deep RL via Discrete Information Bottleneck. Islam et al.
>
> [6] The Information Bottleneck Method. Tishby et al.

---

> > ### Comment · Reviewer_6aXd · 2023-08-16
> > **Post-Rebuttal Comment - Raising Score to a 5**
> >
> > Thanks to the authors for their clarifications! After reading through these and the global comments, as well as the comments from the other reviewers, I feel like many of my original questions around the paper have been addressed.
> >
> > I still believe that the link between the current process of degrading a representation via an entropy penalty and making broader claims about emerging levels of abstraction to be theoretically tenuous, but the user study and responses from the authors raise many good points, downweighting this initial comment from my review. I think many of my issues around the soundness of the paper have been addressed.
> >
> > I'm still not sure about the experiments controlling for confounds that I mention, and the general viability of the new framing of the paper (the human-in-the-loop approach doesn't seems super motivated, for the reasons that Reviewer TMXo stated) either, but I am raising my score to a borderline accept (5).

---

### Author Rebuttal · Authors · 2023-08-04

We thank all reviewers for their comments. We have replied to specific questions in individual responses. Here, we briefly highlight results that address some common themes:

# Regularization baselines:

Reviewers 99tN and TMXo asked for stronger finetuning baselines, with regularized models trained with a validation set. We thank the reviewers for this excellent suggestion.

We implemented this approach, using both L2 and L1 regularization on model weights, sweeping through regularization weights from 0.00001 to 0.1 at powers of 10 (and the default of 0.0) and selected the best weight when assessed on a validation set of 1 example per class. Results from such experiments (using L2 regularization), are included in the attached pdf, for $\beta$-VAE, VQ-VIB$\_\mathcal{C}$, and VQ-VIB$_\mathcal{N}$, in all domains (using $n = 1$ for both VQ-VIB methods, as it performed best). Results were nearly identical for L1 regularization.

Overall, regularization improved finetuning performance of all models by a small amount, but the main trends of our experiments still held. That is, for small $k$, we observed that finetuning performance first increased and then decreased as VQ-VIB$_\mathcal{C}$ models used more abstract representations, confirming the value of our method.

The greatest benefit from using regularized models seemed to be in the lowest MSE range. With a “perfect” regularization method, we might expect that performance would plateau at low MSE rather than worsen. Instead, using L2 regularization, we found that performance decreased slightly as MSE decreased, but performance decreased less than without regularization. This aligns with the intuition that regularization can help for the most complex representations, but consistently finding the right regularization method remains challenging.

Critically, these findings corroborate the main idea of our paper: that compressing representations remains an important mechanism for optimal finetuning performance in low-data regimes, and that end users are adept at selecting the optimal compressed representations adaptively at test time. Upon acceptance, we would update all figures in the main paper to use the regularized baselines.

# Real-world use cases

Reviewers 99tN and TMXo asked for a real-world use case of our method. We acknowledge in our paper that our method confers the largest benefit in low-data visual regimes (we do not claim to solve all transfer learning scenarios), which are well motivated in many real-world tasks such as visuomotor robot learning and end user home personalization, where the ability to collect data personalized to users remains limited [1]. Consider the scenario where a visuomotor policy has been trained to help a user sort clothing in their home or identify mugs for making coffee [2]. In this problem setting, we wish to be able to identify specific types of clothing or mugs that the user uniquely possesses in their home, and adapt our policy to these personalized preferences without the need for the user to provide large-scale training data from their home. Using our approach, we could train a robot to identify clothing for sorting (as in our FashionMNIST example) using only 10 examples per class and achieve roughly 93% accuracy. Moreover, if the user decides to give the robot to their family member, we could adaptively perform the same finetuning process to the new user’s preferences! This type of flexible test-time adaptation would be extremely desirable in these scenarios.

# Autonomously selecting the right representation via prototypes

Reviewers 99tN and TMXo suggested the hypothesis that the optimal encoder is the one that uses approximately the same number of prototypes as finetuning labels. Such an approach does not succeed.

Consider the following theoretical example in FashionMNIST. Using the same pre-trained encoders as in our main paper, we seek to finetune models based on a binary classification task: handbags with straps and handbags without straps. Given that there are two finetuning labels, one might consider encoders that use two prototypes. However, our FashionMNIST encoders that only use two prototypes typically use one prototype to represent shoes and one prototype to represent everything else. Such encoders would fail catastrophically at fine-grained discrimination between types of handbags.

Empirical results corroborate the intuition from this theoretical example.

In the CIFAR100 20-way finetuning task (which contains odd groupings, such as two distinct categories for vehicles) in the main paper, peak performance was at the lowest MSE values, which corresponded to using roughly 100 prototypes. Conversely, using 20 prototypes resulted in worse performance.

Furthermore, we conducted an additional FashionMNIST experiment where models were finetuned on a new 3-way classification task. Groupings were intentionally non-meaningful (Group 1: Tshirt, Trouser, Sandal; Group 2: Pullover, Dress, Sneaker; Group 3: Coat, Shirt, Bag, Boot). In this domain, peak performance was achieved for models using far more than just 3 prototypes. Once again, this establishes that if the fine-grained labels do not align with how VQ-VIB$_\mathcal{C}$ models learn compressed representations, models benefit from having more prototypes (and less compressed representations).

These experiments establish that autonomously selecting the optimal encoder for a finetuning task remains challenging. Conversely, humans are adept at choosing the right model. We conducted an additional online user study using the novel 3-way grouping for FashionMNIST and found that participants were able to correctly identify the optimal abstraction level for this task as well. This reinforces the value of our human-in-the-loop framework.

[1] "Robot learning from randomized simulations: A review." Muratore et al.

[2] "Diagnosis, Feedback, Adaptation: A Human-in-the-Loop Framework for Test-Time Policy Adaptation." Peng et al.

---

### Author Response · Authors · 2023-08-12
**Paper reframing per suggestions!**

# Paper Reframing

We thank reviewers for their excellent comments and active engagement! In light of the discussion and the recommendations suggested by Reviewers 99tN and TMXo, we propose the following reframing of our paper:

1. We explore the hypothesis that pretraining with lower representation complexity is beneficial to downstream finetuning data efficiency. We propose a new architecture, VQ-VIB$_\mathcal{C}$, for generating discrete prototypes with varied representation complexities.

2. We find that, with enough finetuning data ($k$>5 from each class), there are little to no benefits to lower complexity models. Thus, fully autonomous methods for selecting the optimal encoder perform well. However, in low-data regimes (such as the provided robot example or situations where task data can only be provided by human users), there is a benefit to lower complexity representations. Fully autonomous methods *fail* to select the optimal encoder in these cases.

3. To address low-data regimes, we propose a human-in-the-loop framework for selecting the optimal encoder. We demonstrate in a user study that this method is successful at selecting the optimally performing encoder.

Overall, these results suggest that in low-data regimes, incorporating human partners in the finetuning process can outperform fully autonomous methods (although this benefit shrinks when we have access to a large enough dataset).

We would like to emphasize that this proposed reframing would require *no additional experimental data*, but is simply intended to better emphasize the findings of our detailed technical evaluation while more clearly acknowledging the limitations of the human-in-the-loop method.

---

### Decision · Program_Chairs · 2023-09-21

**Decision:**

Accept (poster)

**Comment:**

This paper shows a method for generating reps at different levels of complexity, experiments showing (unfortunately slightly lukewarm) results on the effect of controlling this, and a human study (whose scalability / strength seems to have been a point of slight contention). However, all reviewers agree that it's an interesting and valuable question, that the experiments are thorough, and ways to involve humans in a ML problem is understudied and interesting. All reviewers converged on an accept decision after a lively rebuttal phase.

In the interest of supporting creative approaches trying new methods on important questions (even if the results don't demonstrate a stellar improvement), I'd recommend it be discussed at NeruIPS / acceptance as a poster.